# Red-shifted channelrhodopsin stimulation restores light responses in blind mice, macaque retina, and human retina

Abhishek Sengupta[1,2,3,†,§], Antoine Chaffiol[1,2,3,†], Emilie Macé[1,2,3], Romain Caplette[1,2,3], Mélissa Desrosiers[1,2,3], Maruša Lampič[1,2,3], Valérie Forster[1,2,3], Olivier Marre[1,2,3], John Y Lin[4], José-Alain Sahel[1,2,3,5], Serge Picaud[1,2,3], Deniz Dalkara[1,2,3,**] & Jens Duebel[1,2,3,*]

## Abstract

Targeting the photosensitive ion channel channelrhodopsin-2 (ChR2) to the retinal circuitry downstream of photoreceptors holds promise in treating vision loss caused by retinal degeneration. However, the high intensity of blue light necessary to activate channelrhodopsin-2 exceeds the safety threshold of retinal illumination because of its strong potential to induce photochemical damage. In contrast, the damage potential of red-shifted light is vastly lower than that of blue light. Here, we show that a red-shifted channelrhodopsin (ReaChR), delivered by AAV injections in blind *rd1* mice, enables restoration of light responses at the retinal, cortical, and behavioral levels, using orange light at intensities below the safety threshold for the human retina. We further show that postmortem macaque retinae infected with AAV-ReaChR can respond with spike trains to orange light at safe intensities. Finally, to directly address the question of translatability to human subjects, we demonstrate for the first time, AAV- and lentivirus-mediated optogenetic spike responses in ganglion cells of the postmortem human retina.

**Keywords** channelrhodopsin; optogenetics; primate; retina; vision restoration
**Subject Categories** Genetics, Gene Therapy & Genetic Disease; Neuroscience

## Introduction

Pioneering studies have demonstrated that targeting of microbial opsins, such as channelrhodopsin-2 (ChR2) (Nagel *et al*, 2003), to surviving retinal neurons restores light sensitivity in degenerated retinae, that light-driven information is transmitted to higher visual centers and that visually guided behavior can be elicited (Bi *et al*, 2006; Lagali *et al*, 2008; Thyagarajan *et al*, 2010; Tomita *et al*, 2010; Doroudchi *et al*, 2011). However, a major drawback is that ChR2 requires stimulation with blue light at high intensities, having high risk of inducing photochemical damage in the retina and the retinal pigment epithelium (Ham *et al*, 1978; Chen, 1993; Rozanowska *et al*, 1995; Sparrow *et al*, 2000, 2002; Lamb *et al*, 2001; Wu *et al*, 2006; Hunter *et al*, 2012). Indeed, stimulation with blue light at levels sufficient to activate ChR2 would exceed the safety threshold of artificial radiation permitted to be applied to the human retina (European Commission, 2006; ICNIRP, 2013).

A solution to this problem is the use of red-shifted channelrhodopsin variants. The key point is that the potential of photochemical damage in the human eye has an exponentially decaying correlation with wavelength, implying that even small changes in the wavelength have a strong impact on the safety threshold. For example, by shifting the wavelength from 470 nm (blue light) to 590 nm (orange light), we are permitted to increase the light intensity for stimulation by three orders of magnitude (European Commission, 2006; ICNIRP, 2013), thus making red-shifted channelrhodopsin variants ideal candidates for optogenetic vision restoration.

The first red-shifted channelrhodopsin derived from *Volvox carteri* (VChR1) exhibits peak response at ∼530 nm but shows poor membrane trafficking when expressed in mammalian cells (Zhang *et al*, 2008). Molecular engineering efforts have led to novel mutants with enhanced membrane expression and more red-shifted action spectra (Lin *et al*, 2013; Klapoetke *et al*, 2014; Tomita *et al*, 2014). Here, we use a recently developed variant, called red-activatable ChR (ReaChR), which displays improved membrane trafficking and

1   INSERM, U968, Paris, France
2   Sorbonne Universités, UPMC Univ Paris 06, UMR_S 968, Institut de la Vision, Paris, France
3   CNRS, UMR_7210, Paris, France
4   School of Medicine, University of Tasmania, Hobart, Tasmania, Australia
5   Hôpital des Quinze-Vingts, Paris, France
   *Corresponding author. Tel: +33 6 52 56 24 00; E-mail: jens.duebel@inserm.fr
   **Corresponding author. Tel: +33 1 53 46 25 32; E-mail: deniz.dalkara@gmail.com
   †These authors contributed equally to this work
   §Present address: Unit on Retinal Neurophysiology, National Eye Institute and Graduate Partnerships Program, National Institutes of Health, Bethesda, MD, USA

robust spectral responses up to 600 nm (Lin *et al*, 2013). By using an AAV vector, we targeted ReaChR to retinal ganglion cells (RGCs) in a mouse model of retinitis pigmentosa (*rd1* mouse). Upon illumination with red-shifted wavelengths, at intensities below the safety threshold allowed for the human eye, we observed ReaChR-induced responses in the retina that were relayed to the visual cortex and mediated visually guided behavior. We also demonstrate that ReaChR is functional, when expressed *ex vivo* in the macaque retina and in the human retina.

## Results

We targeted retinal ganglion cells (RGCs) of blind *rd1* mice (4–5 weeks old) with an AAV2 encoding ReaChR-mCitrine under a pan-neuronal hSyn promoter via intravitreal injections. Four weeks post-injection, we observed strong retinal expression of mCitrine in *rd1* mice as revealed by *in vivo* fundus imaging (Fig 1A). Live 2-photon imaging of ReaChR-treated retinae (Fig 1B) as well as confocal imaging (Figs 1C, and EV1A) exhibited endogenous mCitrine fluorescence localized to the RGC membrane as well as dense labeling of RGC dendritic arbors in the inner plexiform layer. Next, we tested the light intensities required to stimulate ReaChR-expressing RGCs (ReaChR-RGCs) in *rd1* mice (> 4 months old). ReaChR-evoked photocurrents in response to different light intensities spanning five logarithmic units were measured with patch-clamp electrodes (Fig 2A). The light sensitivity of ReaChR-expressing RGCs is lower than that of wild-type cones (Nikonov *et al*, 2006) (Fig 2B). However, the light levels required for ReaChR stimulation at 590 nm ($n = 7$ cells) are below the

regulatory limit of safe illumination of the human retina (orange dashed line). For comparison, the safety threshold for blue light illumination at 470 nm, which is normally used to stimulate ChR-2, would be ~1,000× lower (blue dashed line) (European Commission, 2006; ICNIRP, 2013). This highlights the importance of using a red-shifted optogenetic vision restoration strategy (see also Table 1). By stimulating at different wavelengths at a constant light intensity (~$10^{16}$ photons cm$^{-2}$ s$^{-1}$, see also Materials and Methods), we measured the action spectrum as seen from ReaChR-evoked photocurrents of a representative RGC (Fig 2C). The normalized action spectrum (Fig 2D) that was calculated both from patch-clamp (solid line; $n = 8$ cells) and from multi-electrode array recordings (dashed line; $n = 168$ cells) shows that ReaChR-RGCs can be activated at red-shifted wavelengths ranging from 500 to 600 nm. To assess the temporal properties of ReaChR-expressing RGCs, we recorded photocurrents with patch-clamp electrodes using light pulses ranging from 2 to 22 Hz. Despite the rather slow response offset ($\tau_{Off}$) of ReaChR (Fig EV1C) (Lin *et al*, 2013), we consistently observed a persisting periodicity in the current response up to 22 Hz (Fig 2E). Then, we quantified the temporal properties of ReaChR-evoked spike trains from multi-electrode array recordings obtained in response to flicker stimulations (Fig 2F). The RGC-spike discharges could follow stimulation light pulses of increasing frequencies up to 30 Hz. For comparison, the standard cinema movie frame refresh rate is 24 Hz, implying that ReaChR responses of *rd1*-RGCs achieve a sufficient temporal resolution for human vision.

In order for optogenetic therapeutic approaches to be useful to blind patients, responses triggered in the retina must be transmitted to higher visual centers in the brain. To address this, we performed

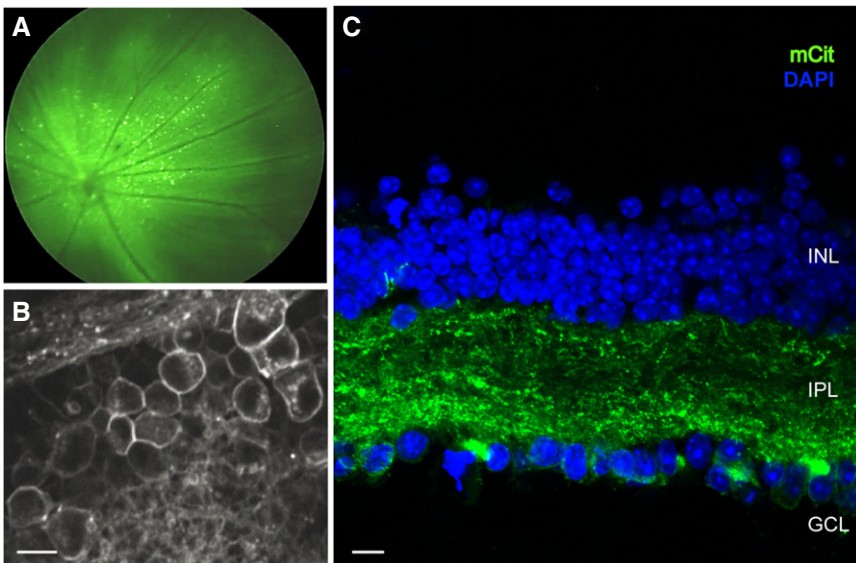

**Figure 1. The red-shifted channelrhodopsin ReaChR can be efficiently targeted to the RGC membrane and dendritic arbor of blind mice.**

A   *In vivo* fundus image of an AAV-injected *rd1* mouse expressing ReaChR-mCitrine driven by the hSyn promoter (AAV2-hSyn:ReaChR-mCitrine); transduced RGCs are visible as green puncta.

B   Live 2-photon image showing that ReaChR-mCitrine expression is restricted to RGC membranes and their dendritic processes. Scale bar, 10 μm.

C   Confocal image of a vertical retinal section expressing ReaChR-mCitrine. Endogenous mCitrine (mCit) fluorescence is localized to RGC membranes as well as RGC dendritic processes (see also Fig EV1A); blue: DAPI (to visualize retinal nuclei); inner nuclear layer (INL); inner plexiform layer (IPL); and ganglion cell layer (GCL). Scale bar, 10 μm.

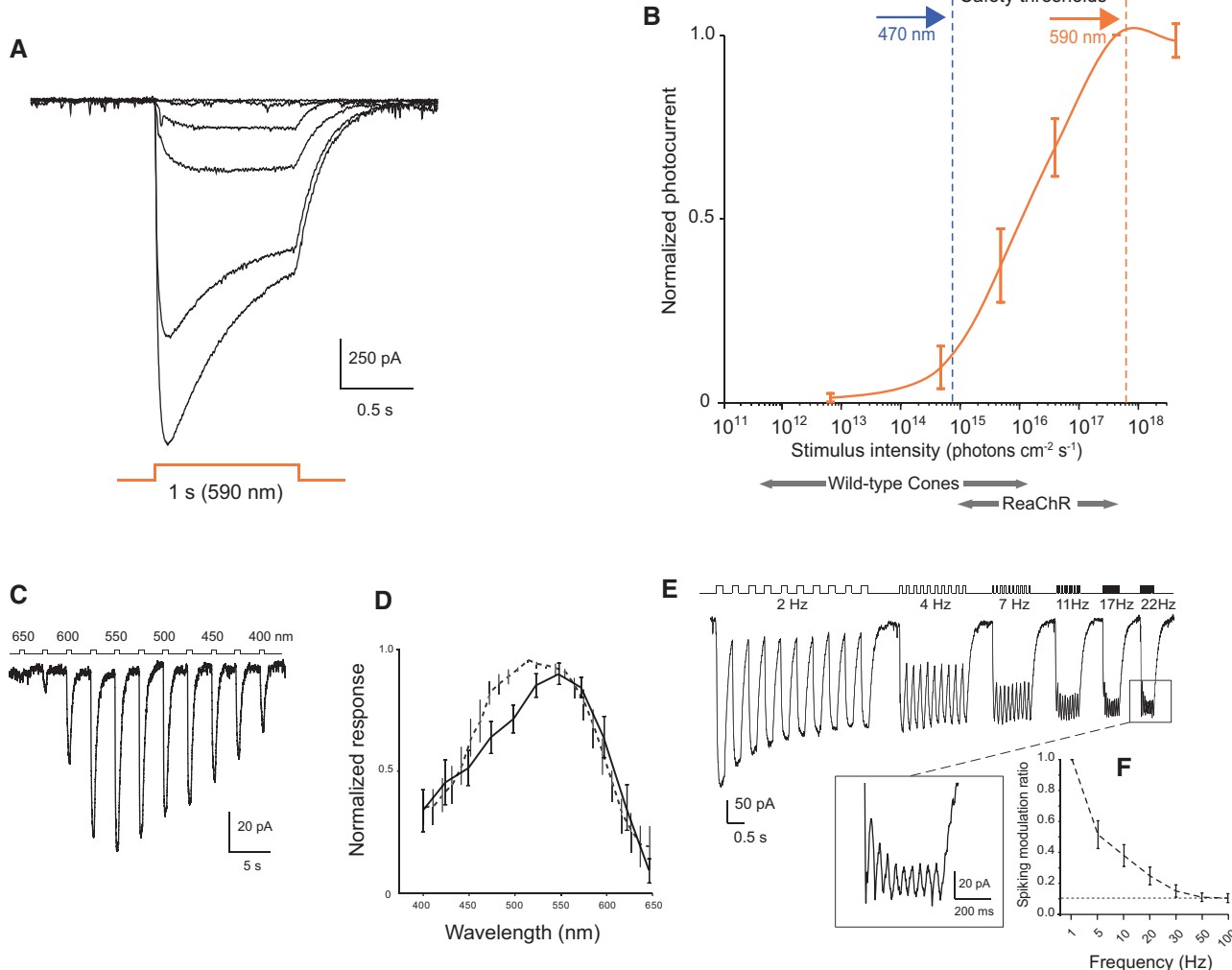

**Figure 2.  Red-shifted optogenetic stimulation of ReaChR triggers responses in the retinae of blind mice injected with AAV2-hSyn:ReaChR-mCitrine.**

A   Light sensitivity: Photocurrents of a ReaChR-expressing RGC stimulated with varying light intensities at 590 nm ($\sim10^{13} - 10^{18}$ photons cm$^{-2}$ s$^{-1}$).

B   ReaChR-induced photocurrents as a function of light intensity (590 nm, $n = 7$ cells, mean photocurrent amplitude at $4.2 \times 10^{17}$ photons cm$^{-2}$ s$^{-1}$ was $-349 \pm 192$ pA). At the bottom, the sensitivity range for wild-type cones (Nikonov *et al*, 2006) is shown in comparison with the sensitivity range of ReaChR-expressing RGCs. The dashed blue and orange lines indicate the maximum light intensities allowed in the human eye at 470 nm and 590 nm, respectively (European Commission, 2006; ICNIRP, 2013).

C   Photocurrent action spectrum of a ReaChR-expressing RGC ($1.2 \times 10^{16}$ photons cm$^{-2}$ s$^{-1}$).

D   Normalized action spectrums of ReaChR-RGCs. Solid line represents patch-clamp data ($n = 8$ cells; $1.2 \times 10^{16}$ photons cm$^{-2}$ s$^{-1}$). The mean photocurrent amplitude at 550 nm was $-324 \pm 97$ pA. Dashed line represents multi-electrode array data ($n = 168$ cells; $\sim10^{16}$ photons cm$^{-2}$ s$^{-1}$).

E   Temporal properties. Modulation of ReaChR-induced photocurrents at increasing stimulation frequencies (patch-clamp recording from a ReaChR-expressing RGC; at 22 Hz, magnified trace is shown; $1.3 \times 10^{16}$ photons cm$^{-2}$ s$^{-1}$).

F   Spiking modulation ratio of ReaChR-RGCs quantified from multi-electrode array demonstrates reliable spike triggering at frequencies up to 30 Hz (297 cells, $1.3 \times 10^{16}$ photons cm$^{-2}$ s$^{-1}$).

Data information: Graphs from (B), (D), and (F) are presented as mean $\pm$ SEM.

electrophysiological recordings in the visual cortex of *rd1* treated with AAV2-hSyn:ReaChR-mCitrine and age-matched control *rd1* mice and wild-type (WT) mice (Fig 3A–C). The ReaChR-treated eye was stimulated with 200 ms orange light (595 nm) pulses. The maximal local field potential (LFP) amplitude in ReaChR-treated *rd1* mice was $-281.2 \pm 68.4$ µV; $n = 4$ (Fig 3A) but flat-lined in untreated *rd1* mice ($-10.3 \pm 2.9$ µV, $n = 3$; *rd1* vs. ReaChR-treated $P < 0.03$, two-tailed *t*-test) (Fig 3A). A representative cortical response profile of a WT mouse stimulated similarly at 595 nm is provided for comparison, in which we observed ON and OFF responses, while the ReaChR-treated *rd1* mice displayed pure ON responses. Additionally, light stimuli produced spike discharges in ReaChR-treated *rd1* mice (Fig 3B) but failed to generate such responses in blind untreated *rd1* mice (Fig 3B). We assessed the sensitivity of cortical responses to light onsets (Fig 3D) in ReaChR-treated blind mice using intensities ranging from $2.5 \times 10^{13}$ to $2.5 \times 10^{17}$ photons cm$^{-2}$ s$^{-1}$ and observed modulations of the LFP amplitude as well as spike discharges beginning at

**Table 1. Safety thresholds for specific wavelengths of light (ICNIRP, 2013).**

| Wavelength: $\lambda$ (nm) | Spectral weighting factor: $B_\lambda$ | Light intensity allowed on the retina: (photons cm$^{-2}$ s$^{-1}$) |
|---|---|---|
| 440 | 1.0 | $\leq 4.43 \times 10^{14}$ |
| 470 | 0.62 | $\leq 7.62 \times 10^{14}$ |
| 500 | 0.1 | $\leq 5.03 \times 10^{15}$ |
| 550 | 0.01 | $\leq 5.53 \times 10^{16}$ |
| 590 | 0.001 | $\leq 5.94 \times 10^{17}$ |

$2.5 \times 10^{15}$ photons cm$^{-2}$ s$^{-1}$, suggesting that the ReaChR-treated blind mice can detect light in their visual cortex at intensities well below the safety threshold of stimulating light.

Could the behavior of the blind *rd1* mice, treated with ReaChR, be modulated by orange light? To investigate this, we placed the mice in a circular open-field like arena fitted with 590-nm LED light sources. The *rd1* mice treated with AAV2-hSyn:ReaChR-mCitrine showed light responses (Fig 4A ReaChR-treated *rd1*, Movie EV1) characterized by a sharp reduction in their velocity of movement with respect to the *rd1* mice ($2.18 \pm 0.33$ cm s$^{-1}$, $n = 11$ ReaChR-treated and $3.55 \pm 0.37$ cm s$^{-1}$, $n = 12$ *rd1*; $P = 0.006$, one-tailed *t*-test) during 2 min after light onset (Fig 4B). The behavior of control age-matched *rd1* mice remained unaffected upon open-field illumination (Fig 4A *rd1*, Movie EV2). Since only ON light responses are elicited in the ReaChR-treated *rd1* mice in our approach, we asked whether these mice could discriminate light and dark fields. Therefore, we used the light/dark box, illuminated with 590 nm light at different intensities ranging from $2.1 \times 10^{13}$ to $2.1 \times 10^{16}$ photons cm$^{-2}$ s$^{-1}$. ReaChR-treated *rd1* mice exhibited robust light avoidance responses, starting at $2.11 \times 10^{15}$ photons cm$^{-2}$ s$^{-1}$, for which the mice spent only $14.43 \pm 4.4\%$ of the total duration of the test in the light compartment (Fig 4C, see Movie EV3). The WT mice spent significantly less time in the illuminated compartment at all light intensities, and their performance increased when the light intensity was elevated. In contrast, the untreated *rd1* mice did not show any light-induced avoidance behavior. Independent from the light intensity, *rd1* mice spent on average ~59% of their time in the illuminated compartment at all the intensities tested (Fig 4C). Fig 4D shows the performance of the animals at $2.11 \times 10^{15}$ photons cm$^{-2}$ s$^{-1}$. At this intensity, the *rd1* control mice spent $59.3 \pm 2.8\%$ ($n = 9$), wild-type mice spent $37.7 \pm 2.44\%$ ($n = 9$), and ReaChR-treated mice spent $14.4 \pm 4.4\%$ ($n = 7$) of their time in the light compartment. Both the wild-type and ReaChR-treated mice spent significantly less time in the light compartment than the *rd1* mice ($P < 0.0001$, one-way ANOVA followed by Tukey's multiple comparisons test to compare *rd1* vs. WT and *rd1* vs. ReaChR-treated).

To evaluate the clinical translatability of our approach, we tested whether ReaChR can be functional, when expressed with an AAV in RGCs of the primate retina. We prepared explants from the mid-peripheral parts of postmortem macaque retinae and infected them with an AAV-ReaChR in tissue culture (Fradot *et al*, 2011). After 10 days of incubation with AAV2-hSyn:ReaChR-mCit, we observed strong membrane-bound fluorescence in RGCs and in their dendrites stratifying in the inner plexiform layer (Fig 5A). In repeated electro-physiological recording trials, we attempted but could not record ReaChR-triggered light responses from explants infected with the

hSyn ReaChR construct. In order to induce stronger expression of ReaChR with the AAV vector under culture conditions, we utilized the ubiquitous CAG promoter instead of the pan-neuronal hSyn promoter and packaged it in AAV8 Y447 733. After 3 days of incubation with AAV8 Y447 733-CAG:ReaChR-GFP, we could detect GFP fluorescence in these explants (Fig 5B) and we were able to record light-induced spike responses upon stimulation with orange light (Fig 5C). To visualize the response characteristics of the treated macaque retinae, we applied 2-s light pulses at 580 nm and recorded the spike responses with a multi-electrode array. Upon light stimulation, the cells displayed sustained responses (Figs 5C and EV2A), demonstrating that light responses can be evoked at a light intensity below the safety threshold allowed in the human eye (see Materials and Methods and Table 1). To reconstruct the red-shifted action spectrum of ReaChR from a larger cell population, we

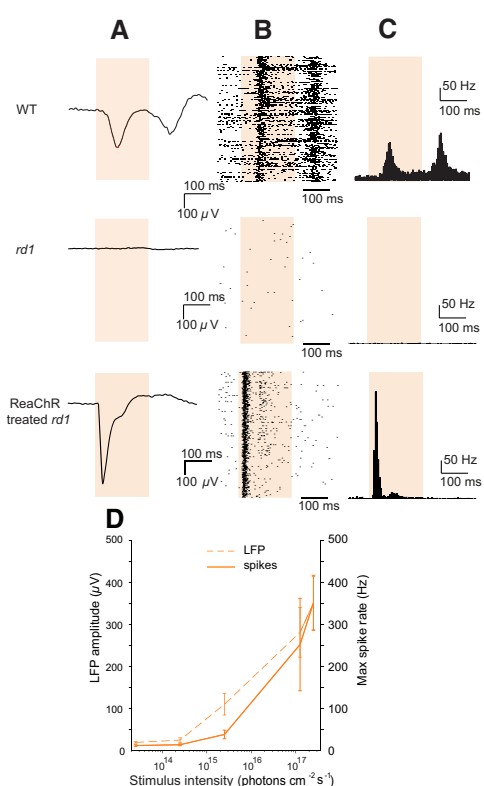

**Figure 3. ReaChR-triggered responses in blind *rd1* mice, treated with AAV2-hSyn:ReaChR-mCitrine, can be detected in the visual cortex at a range of orange light intensities below the safety threshold.**

A   Examples of local field potential (LFP) traces in response to light flashes at 595 nm (light stimulus is shown as orange bar; $1.25 \times 10^{17}$ photons cm$^{-2}$ s$^{-1}$).

B   Spike raster plots in response to 200 repetitions of the same light stimulus.

C   Averaged spike responses shown as peristimulus time histograms (PSTH, bin size: 5 ms) based on the spike raster plots. Top row: WT mouse, middle row: *rd1* control mouse, bottom row: ReaChR-treated *rd1* mouse.

D   ReaChR-induced cortical responses (spikes and LFP) as a function of light intensity at (595 nm) demonstrating that ReaChR-treated *rd1* mice ($n = 4$) can detect light in the visual cortex at intensities well below the safety threshold allowed for the human retina. Data presented as mean $\pm$ SEM.

Data information: Plots in (A–C) have been constructed from responses of a single representative WT, *rd1* and ReaChR-treated *rd1* mouse, respectively. Plots in (B and C) are representative single-unit neural responses.

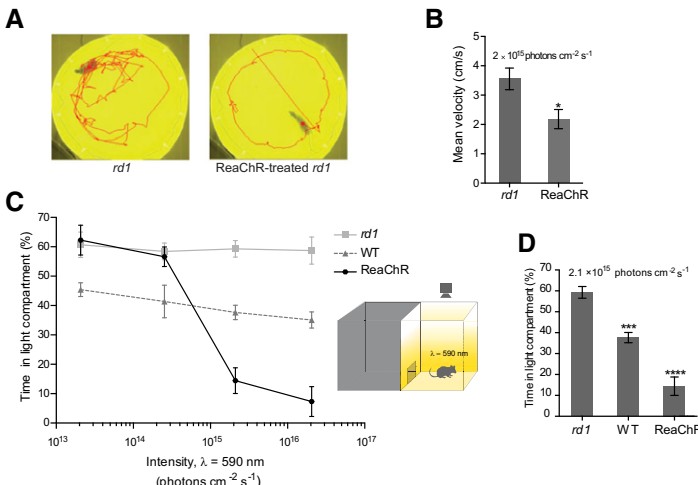

**Figure 4.  ReaChR-evoked responses are sufficient to modulate the behavior of blind mice treated with AAV2-hSyn:ReaChR-mCitrine.**

A  Locomotory behavior of a representative *rd1* and ReaChR-treated mouse during illumination with light at 590 nm in an open-field arena (irradiance measured at the level of the mice's eyes was ~$10^{15}$ photons cm$^{-2}$ s$^{-1}$; cumulative activity trajectories are shown as red traces after 2 min of illumination). See also Movies EV1 and EV2.

B  Mean velocities of the ReaChR-treated and *rd1* mice analyzed over a duration of 2 min at light onset. The mean velocity of ReaChR-treated mice was significantly reduced compared to *rd1* mice (*$P$ = 0.006, one-tailed *t*-test, *rd1* vs. ReaChR) upon illumination. Mean ± SEM are shown for each group. ReaChR-treated *rd1* ($n$ = 11), *rd1* ($n$ = 12).

C  Light/dark box test. Percentage of time mice spent in the light compartment as a function of light intensity (WT, $n$ = 9; *rd1*, $n$ = 9; ReaChR-treated, $n$ = 7). Mean ± SEM are shown for each group.

D  Percentage of time *rd1* ($n$ = 9), WT ($n$ = 9) and ReaChR-treated ($n$ = 7) mice spent in the light compartment at an illumination intensity of 2.11 × $10^{15}$ photons cm$^{-2}$ s$^{-1}$. Both WT and ReaChR-treated mice spent significantly less time in the light compartment than the *rd1* mice ($P$ < 0.0001, ordinary one-way ANOVA). ***$P$ < 0.001 and ****$P$ < 0.0001 indicate significance levels determined by Tukey's multiple comparisons test for *rd1* vs. WT and *rd1* vs. ReaChR-treated, respectively. See also Movie EV3 for the response of a representative ReaChR-treated *rd1* mouse.

averaged the data from three AAV-infected macaque retinae from three animals (28 cells) and observed a peak at ~550 nm and light responses persisting at more red-shifted wavelengths up to 600 nm (Fig 5D). Finally, we evaluated the temporal properties of ReaChR-RGCs in the primate retina. In cell-attached patch-clamp recording, we observed that ReaChR-evoked spike responses in the macaque retina could follow stimulus frequencies up to 22 Hz (Fig 5E). All electrophysiological experiments were performed with bath application of L-AP4 to block possible remaining input from the photoreceptors. Control multielectrode array experiments performed on uninfected macaque retinal explants (three retinae from two animals) in the presence of L-AP4 showed complete abolition of endogenous ON light responses and only spontaneous spiking activity remained (see Fig EV2C for representative voltage traces).

To directly address the issue of translatability to human subjects, we took advantage of postmortem human retinae obtained from donor eyes. We infected an explant from a human retina, covering approximately half of the macula, with a lentivirus encoding ReaChR-mCitrine under hSyn promoter (Fig 6A). A lentivirus was chosen because the lentiviral vector has been shown to induce rapid gene expression driven by tissue-specific promoters even under postmortem culture conditions (Busskamp *et al*, 2010). This allowed us to test whether the same promoter–opsin combination (hSyn:ReaChR), that we used in the mouse, is functional in the human retina as well. After 4 days of incubation in tissue culture, we performed electrophysiology on the human retinal explant such that the fovea, as well as the para-foveal region, was put in contact with the multi-electrode array (Fig 6B). We observed spontaneous

spiking activity in the para-foveal area and, as expected, a total absence of spiking in the "photoreceptor-only" fovea (Fig EV3A and B). Stimulation at 600 nm induced spike responses, demonstrating the functionality of ReaChR in the human retina (Fig 6C). Note that the human retina was devoid of innate light responses at the time of isolation. However, to rule out any potentially remaining synaptic input from photoreceptors, the recordings were performed in Ames medium containing 50 μM L-AP4 after 20 min of incubation in the drug. Then, we used light stimuli at wavelengths ranging from 400 to 650 nm. To visualize the ReaChR-evoked spike trains in the human retina, we constructed peristimulus time histograms (PSTHs) and raster plots of the ReaChR responses at different wavelengths, demonstrating that ReaChR can trigger sustained spike trains in human RGCs with safe red-shifted stimuli up to 600 nm (Fig 6E). This response profile (Fig 6D and E) matched the action spectrum of ReaChR, which we also observed in the retinae of the blind *rd1* mice (Fig 2C and D) as well as the macaque retinal explants (Fig 5D). Finally, with confocal microscopy, we examined the anatomy of ReaChR-expressing RGCs in the para-fovea (Fig 7A) and in the near periphery (Fig 7B). Fig 7A shows ReaChR expression in para-foveal RGCs, located within ~500 μm from the foveal pit, which was the same region from where we recorded ReaChR-triggered light responses. In this region of the human retina, the density of midget GCs is estimated to be ~95% of the total GC population (Dacey, 1993). Although the morphology of these cells could be best studied by dye injection techniques, based on the soma sizes of the fluorescent cells as well as their close proximity to the fovea (Dacey & Petersen, 1992; Dacey, 1993), we speculate that the cells

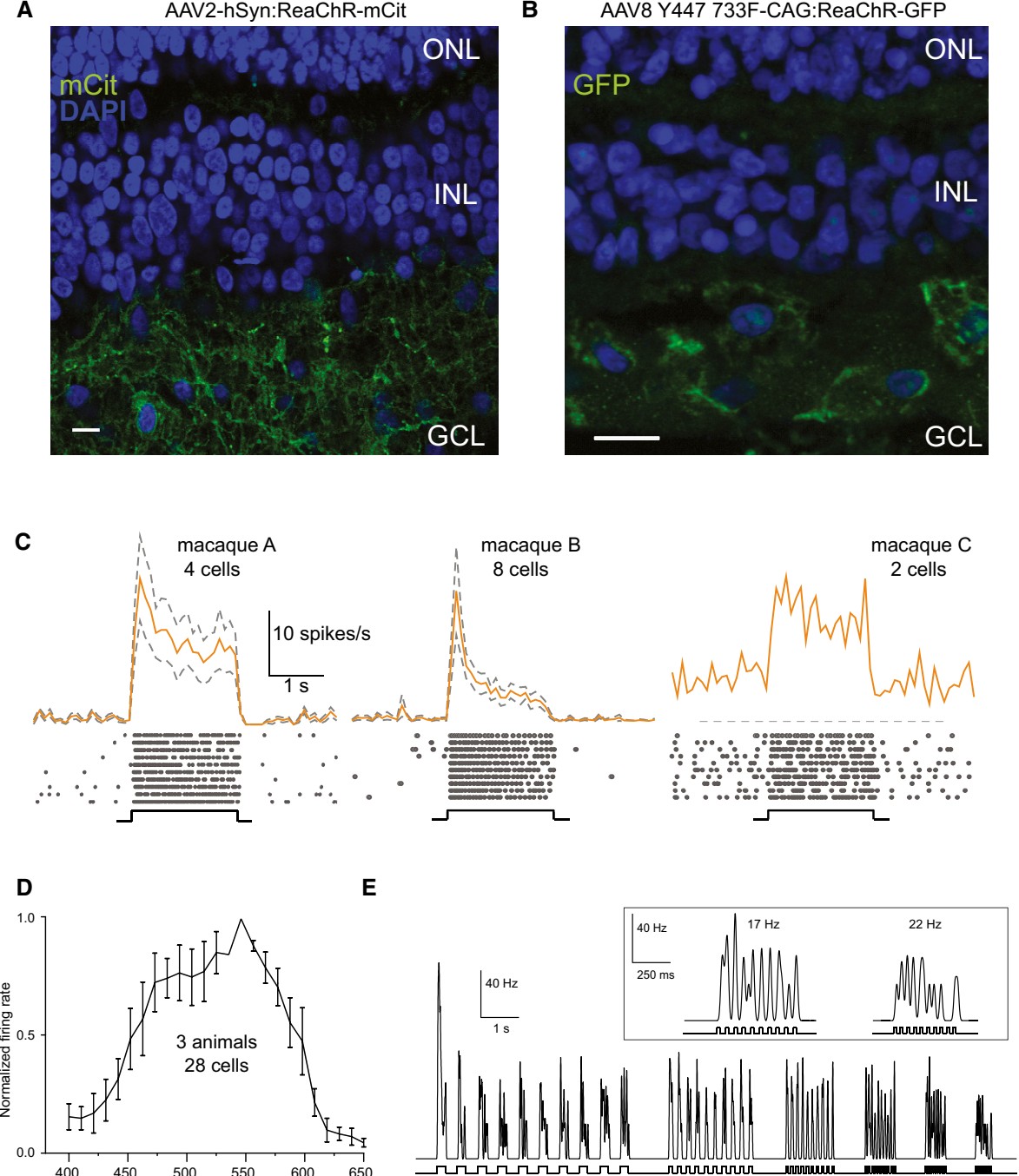

**Figure 5. Evaluation of the translational potential of ReaChR expression and light responses in AAV-infected macaque retinae.**

A   Vertical section of macaque retinal explant transfected with AAV2-hSyn:ReaChR-mCit after 10 days of incubation exhibiting mCitrine expression densely in neuronal processes in the GCL. Scale bar, 10 μm.

B   Vertical section of macaque retinal explant transfected with AAV8 Y447 733F-CAG:ReaChR-GFP after 3 days of incubation. GFP expression was primarily observed in the cell bodies but also in processes of RGCs; green: endogenous GFP fluorescence; blue: DAPI; outer nuclear layer (ONL). Scale bar, 10 μm.

C   Averaged spike responses obtained from multi-electrode array recordings shown as PSTH of 4, 8, and 2 cells, respectively, of three different macaques (580 nm; $1.24 \times 10^{17}$ photons $cm^{-2}$ $s^{-1}$). PSTHs are presented as mean $\pm$ SEM (dashed lines, calculated over responding cells). Raster plots of a representative cell from each individual macaque are shown below (10 stimulus repetitions).

D   Action spectrum of ReaChR in primate RGCs from multi-electrode array recordings. Data are shown as mean $\pm$ SEM of the normalized firing rate, averaged from three individual macaques (28 cells, error bars calculated over cells), see also Fig EV2B for action spectrums of three individual macaques obtained from multi-electrode array recordings.

E   Responses (Hz) of a ReaChR-expressing RGC (patch-clamp recording, in cell-attached mode) to flicker stimulations at increasing frequencies (2–22 Hz; $1.3 \times 10^{16}$ photons $cm^{-2}$ $s^{-1}$).

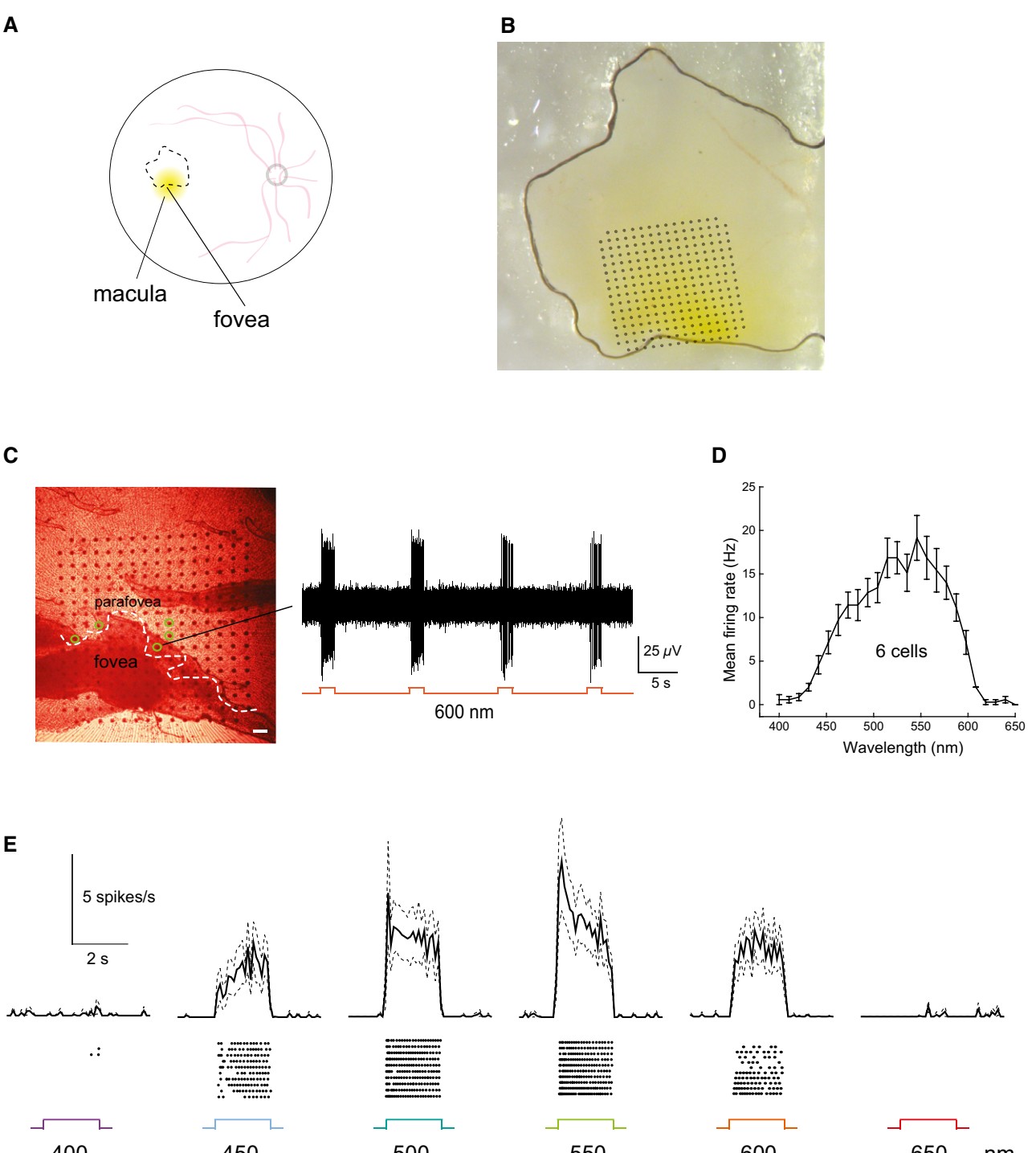

**Figure 6.  In the human retina, ReaChR is functional in ganglion cells of the para-fovea at light intensities safe for the human eye in a macular explant transduced with lentivirus encoding hSyn:ReaChR-mCitrine.**

A  Schematic illustrates the size and location of the retinal explant (dashed line).
B  Brightfield image of the retinal explant showing the macula (yellow pigmented) with the approximate location of the multi-electrode array chip indicated.
C  Infrared image of the explant pressed against the multi-electrode array; the white dashed line depicts the "electrophysiological boundary" of the para-fovea based on the absence of RGC-spike trains (scale bar, 100 μm; see also Fig EV3A and B). Voltage trace of a para-foveal RGC stimulated with orange light (600 nm; $1.1 \times 10^{17}$ photons $cm^{-2}\ s^{-1}$) is shown on the right. Individual electrodes that recorded ReaChR-responding cells are indicated in green.
D  Action spectrum of ReaChR constructed from 6 cells (mean ± SEM, error bars calculated over cells) exhibiting peak activity at 550 nm ($1.1 \times 10^{17}$ photons $cm^{-2}\ s^{-1}$) similar to what was observed in mice and in the macaque retinae.
E  Peristimulus time histograms constructed for wavelengths 400–650 nm ($1.1 \times 10^{17}$ photons $cm^{-2}\ s^{-1}$) demonstrate ReaChR-evoked spike trains (under L-AP4) in the human retina (6 cells, mean ± SEM, error bars calculated over cells, bin size: 100 ms). Bottom: Raster plots of the light responses of a representative spike-sorted cell.

transduced in the para-fovea of this human explant could include midget GCs. At a distance of 3–4 mm from the fovea (Fig 7B), we observed ReaChR-expressing RGCs with larger dendritic arborizations than the ones in the para-fovea, which is consistent with the finding that RGC dendritic field sizes increase with retinal eccentricity (Dacey & Petersen, 1992). Importantly, we determined that the endogenous fluorescence was targeted to the RGC membranes as well as dendritic processes (Fig 7A and B), confirming that ReaChR displays similar properties in mouse and human tissue in terms of its membrane trafficking efficiency.

To investigate whether ReaChR could be expressed in the human retina using AAV, we prepared explants from far peripheral parts of the retina and infected them with an AAV2-hSyn:ReaChR-mCit ($2.5 \times 10^{13}$ vector genomes ml$^{-1}$) as well as with an AAV8 Y447 733-CAG:ReaChR-GFP ($6.3 \times 10^{13}$ vector genomes ml$^{-1}$). After 9 days of viral infection, we observed very weak expression of the hSyn construct (AAV2 capsid) but a higher level of expression of the CAG construct (AAV8 Y447 733 capsid) in a sparse population of RGCs (Figs 8 and EV4). We estimate that the ubiquitous promoter as well as the AAV8 Y447 733 capsid could have played a role in the stronger expression of the latter construct *ex vivo*. At 12 days post-infection, a retinal explant prepared from the far peripheral human retina, exhibiting ReaChR-GFP expression driven by CAG promoter (AAV8 Y447 733 capsid), was mounted on a multi-electrode chip for electrophysiological recordings (Fig 8A). Stimulation with light pulses at 600 nm elicited ReaChR-mediated spike responses under L-AP4 blockade (Fig 8B). The light response characteristics of responding cells included distinct transient components (Fig 8B), compared to cells expressing ReaChR in the human para-fovea (Figs 6C and E, and EV3B). Stimulation at different wavelengths

showed that the responses of these cells peaked at 550 nm and were also elicited at 600 nm (Fig 8C).

## Discussion

We have demonstrated that ReaChR can restore light responses in blind mice, in the retina of macaques, as well as in the human retina. Using red-shifted wavelength, we were able to induce robust responses in RGCs at light intensities that are safe for the human eye. Concerning the light safety standards, we refer to the European Directive on artificial optical radiation (European Commission, 2006) and the guidelines of the International Commission on Non-ionizing Radiation Protection (ICNIRP) (ICNIRP, 2013). It is important to note that these safety standards have been designed to protect a healthy retina from light-induced damage. Despite these stringent standards, our results show that the light intensities needed to activate ReaChR are well below the safety threshold when we use red-shifted stimulation (580–600 nm).

The strategy of expressing ReaChR in RGCs could be a therapeutic option for blind patients with severe retinal degeneration who display highly disorganized bipolar cell layers (Jacobson *et al*, 2013; Jones *et al*, 2016). Indeed, for those patients, the only option would be to target RGCs. A drawback of this approach is that the retinal ON/OFF pathway cannot be restored, because all RGCs are turned into ON cells. The development of specific promoters that would allow for targeting ON-RGCs with ChR-2, as well as targeting OFF-RGCs with hyperpolarizing opsins, could help to restore a more naturalistic ON/OFF circuitry. On the other hand, clinical results from RGC stimulation with epiretinal implants indicate that the human cortex has the capability to adapt to a visual code consisting of only ON-RGC responses, that is different from the neural activity pattern conveyed by a healthy retina, and that patients with these implants can perform visual tasks, such as object localization, motion discrimination, and discrimination of oriented gratings (Humayun *et al*, 2012; Shepherd *et al*, 2013). In general, one should keep in mind that the choice of cell type depends on the availability of surviving retinal cells. Hence, patients with an intact inner nuclear layer and viable synapses between bipolar and ganglion cells could be eligible candidates for a bipolar cell-based optogenetic treatment with ReaChR (Jacobson *et al*, 2013; Jones *et al*, 2016). The feasibility of targeting channelrhodopsin variants specifically to ON bipolar cells via AAV vectors has been demonstrated in mouse models of retinal degeneration (Doroudchi *et al*, 2011; Cronin *et al*, 2014; Mace *et al*, 2015), but this has not yet been accomplished in non-human primates.

What other optogenetic tools could be used to restore vision instead of microbial opsins? An alternative strategy could be the use of vertebrate opsins, such as melanopsin or rhodopsin. Vertebrate opsins display higher sensitivity because they amplify the light stimulus by G protein-coupled signaling cascades (Herlitze & Landmesser, 2007). Previous studies have shown that melanopsin—when expressed directly in retinal ganglion cells (Lin *et al*, 2008) or as a melanopsin-mGluR6 chimera in bipolar cells (van Wyk *et al*, 2015)—can be activated at low light levels and that rhodopsin can serve as a highly light-sensitive vision restoration tool in bipolar cells (Cehajic-Kapetanovic *et al*, 2015; Gaub *et al*, 2015). However, it is important to remember that the slow kinetics of melanopsin leads to poor temporal resolution and that rhodopsin is prone to

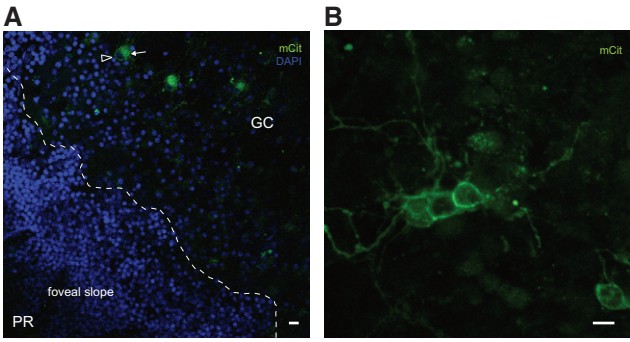

**Figure 7. In the human retina, ReaChR was expressed in para-foveal RGCs as well as in near-peripheral RGCs when a macular explant was transduced with a lentivirus encoding hSyn:ReaChR-mCitrine.**

A   Confocal image of the central retina, exhibiting endogenous ReaChR-mCitrine fluorescence (mCit) in ganglion cells, located within ~500 µm from the center of the foveal pit, a part of the human retina where midget cell density is estimated to be ~95% of total RGC population (Dacey, 1993) (arrowhead: soma; arrow: dendritic arbor). The dashed line illustrates the beginning of the foveal slope leading toward the "cone-only" foveal pit.

B   Confocal image of the same explant at a higher retinal eccentricity (3–4 mm from the foveal center) showing ReaChR-mCitrine fluorescence in RGCs with larger dendritic arbors. The endogenous mCitrine was not amplified.

Data information: Scale bars, 10 µm. PR: photoreceptors, GC: ganglion cells, DAPI (blue).

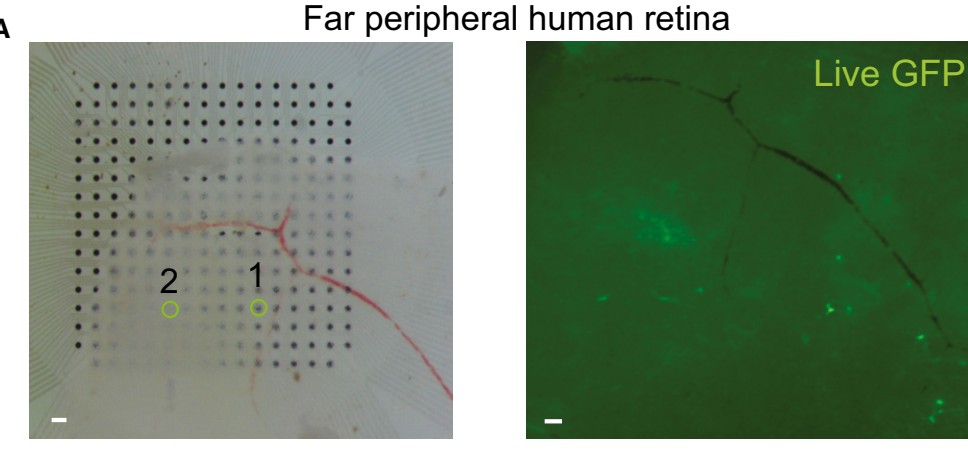

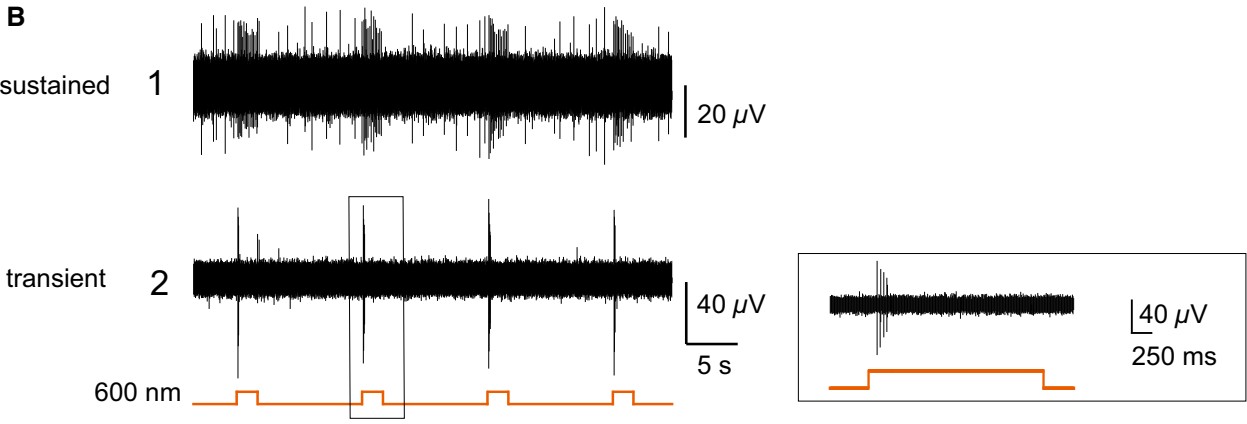

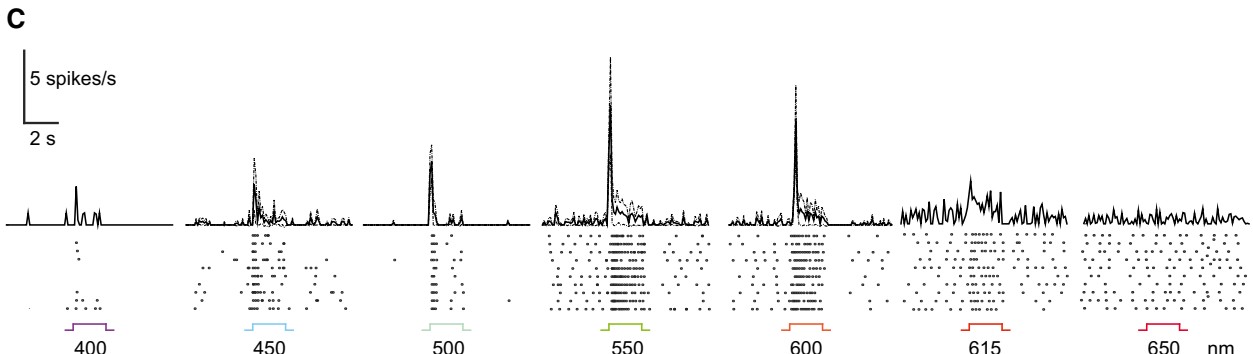

**Figure 8.    AAV-mediated optogenetic activation of the human retina.**

A    Human retinal explant, prepared from the far periphery, infected with an AAV8 Y447 733F-CAG:ReaChR-GFP, placed on a multi-electrode array chip (left); GFP epifluorescence of the same explant following 12 days of incubation is shown on the right. Scale bars, 100 μm. The location of two cells that responded to repeated 2 s full-field stimulations of orange light (600 nm) is highlighted.

B    The corresponding voltage traces show that the responses in the far peripheral human retina are more transient, compared to the experiments from the human para-fovea (see Figs 6C and E, and EV3B) where only sustained light responses were evoked. The light response of cell (2) was restricted to a burst of action potentials fired within the initial ~300 ms in response to a 2 s full-field 600-nm light stimulus.

C    Peristimulus time histograms constructed for wavelengths from 400 to 650 nm ($1.1 \times 10^{17}$ photons $cm^{-2}$ $s^{-1}$) demonstrate AAV-ReaChR-evoked spike trains (mean ± SEM, error bars calculated over 3–4 cells, bin size: 100 ms). Bottom: Raster plots of the light responses of a representative spike-sorted cell. All recordings were performed under pharmacological blockade of photoreceptor input (50 μM L-AP4).

bleaching. While the temporal property of melanopsin has been improved through molecular engineering (Opto-mGluR6) toward faster kinetics (van Wyk *et al*, 2015), bleaching of rhodopsin remains an unsolved problem. When rhodopsin is used as an optogenetic tool in bipolar cells or RGCs, the question remains how continual chromophore recycling will be accomplished in order to provide enough 11-*cis* retinal, which is necessary for phototransduction. In general, the close contact of the retina to the retinal pigment epithelium (RPE) is considered to be important for the visual cycle (Strauss, 2005). This might be a problem for the use of rhodopsin, particularly in retinitis pigmentosa patients with photoreceptor degeneration or in patients with RPE dysfunction, which is often the case with atrophic macular degeneration. As Müller glial cells have been identified as an alternative source of retinal recycling for cone photoreceptors (Kaylor *et al*, 2013), it would be interesting to investigate whether bipolar cells or RGCs have access to this biochemical pathway of chromophore replenishment. In contrast, microbial opsins have the advantage to not bleach because their retinal stays covalently bound after photo-isomerization, and it relaxes spontaneously to its light-sensitive isoform (all-*trans*) in the dark, thus providing the regeneration of the chromophore. Melanopsin and Opto-mGluR6 also have the advantage that they are resistant to bleaching (Sexton *et al*, 2012; van Wyk *et al*, 2015).

The light intensity required to elicit ReaChR-mediated responses on the behavioral level was $10^{15}$ photons cm$^{-2}$ s$^{-1}$, which corresponds to bright indoor light conditions. It is important to note that the ability of the ReaChR-treated mice to discriminate light fields from dark fields was tested but not their ability to perform sophisticated visual tasks. We think that visual acuity and other higher visual tasks in an RGC-based optogenetic approach would be best studied with behavioral tests in AAV-injected non-human primates as a next step. A critical point of ReaChR-expressing RGCs is their inability to adapt to different light intensities. This is consistent with the data from our light/dark box test, where the ReaChR-treated mice showed no response at low light levels ($<10^{14}$ photons cm$^{-2}$ s$^{-1}$), but they exhibited a robust response at high intensities ($>10^{15}$ photons cm$^{-2}$ s$^{-1}$). At these high intensities, their behavioral response was even stronger than the response of WT-mice, which can be most likely explained by the absence of adaptation in the ReaChR-treated mice (Demb, 2008). Therefore, in a future clinical application, stimulation glasses would be necessary in order to stimulate ReaChR with appropriate light intensity. These eyeglasses could use a camera that allows adjusting to a wide range of intensities, and the images captured by this camera could be projected via micro LED arrays onto the retina in order to activate ReaChR at a safe wavelength of 590 nm.

By testing the functionality of ReaChR in the primate retina, we evaluated the translational potential of our approach at the cellular level. In our macaque explant experiments, we recorded from retinae within a 3- to 4-day time-window post-dissection. The sensitivity of optogenetic responses was lower than what we observed from retinae isolated from *in vivo* injected mice, even though responses could still be triggered below the safety threshold of the stimulation light. Our findings provide preliminary evidence that ReaChR could be functional in the primate retina at safe light intensities. We expect that these results would have been significantly improved, in terms of light sensitivity, if recordings were to be performed on freshly isolated retinae with high-density para-foveal expression of

ReaChR, that could be obtained from *in vivo* AAV-treated macaques. Since high-level expression of a microbial opsin may elicit cellular immune responses in the human eye, tolerability of ReaChR has to be evaluated carefully by *in vivo* studies in non-human primates.

The unique morphological and physiological characteristics of the human retina prompted us to investigate functional aspects of our approach directly in the human retina. We expressed ReaChR in RGCs of the postmortem human retina, where we demonstrated it to be functional at red-shifted wavelengths. These results demonstrate, for the first time, that it is possible to trigger spike responses in the human retina with an optogenetic technique. We investigated ReaChR-triggered responses of RGCs directly in the para-fovea as well as those of RGCs in the far peripheral retina. In the intact human retina, the foveal cone signals are carried by midget RGCs, characterized by their minute dendritic arbors (5–10 μm) that underlie the encoding of high spatial information (Rodieck *et al*, 1985; Dacey & Petersen, 1992; Dacey, 1993; Rodieck & Watanabe, 1993). These midget cells dominate the central retina especially near the foveal center but are also present in the peripheral retina where their dendritic fields are larger (Dacey, 1993). A key advantage of optogenetics is that it allows cellular level control of neural activity at high spatial resolution. This should facilitate the stimulation of RGCs with very small receptive fields in the para-fovea. Existing retinal implants that are known to restore visual responses in patients have been explored recently for high-acuity electrical stimulation (Sekirnjak *et al*, 2008; Jepson *et al*, 2014; Lorach *et al*, 2015). However, even with high-density electrodes, stimulation of very small receptive fields with a size of ca. 10 μm would not be accomplished. Here, we show that we can induce optogenetic responses in RGCs immediately surrounding the fovea, and expect that it would be possible to stimulate these small dendritic fields, exploiting the central human retina's innate high-acuity midget RGC circuitry. Although the ganglion cell-targeted optogenetic approach, in principle, would allow stimulation of individual cells without activation of other cells, this issue should be further investigated in the para-fovea. In the para-fovea, where densely packed RGCs form several layers, there exists the possibility of simultaneous stimulation of multiple cells at different layers, risking the distortion of the neural code. In our work, we optogenetically stimulated and recorded from small collections of RGCs in the para-fovea and in the far peripheral human retina. The *ex vivo* viral labeling of the human retina was sparse, and morphological examination suggests that we did not label RGCs stacked in the same vertical column in the para-fovea. Future studies, especially *in vivo* injected primate retinae with high-density expression of optogenetic tools in the para-foveal RGCs, could provide the means to address the computational implications of the simultaneous stimulation of RGCs at different layers in the para-fovea. Further optimization of our *ex vivo* culture system to improve the number of cells expressing optogenetic protein would allow for addressing such questions in the human retina.

In a healthy retina, temporal characteristics of RGCs, such as transient/sustained light responses, are modulated by intrinsic properties of the ganglion cell membranes as well as synaptic inputs from inner retinal circuits (Thyagarajan *et al*, 2010). A therapeutic approach using direct optogenetic stimulation of RGCs would lack such synaptic input from the inner retina. When we recorded ReaChR-triggered light responses from human para-foveal RGCs, we found that their firing patterns were entirely sustained, consistent with the sustained

response patterns expected of midget RGCs' intrinsic light responses. However, in far peripheral human RGCs, we observed distinct transient components in their optogenetic light responses. Our results suggest that the intrinsic properties of RGC membranes can shape their sustained/transient optogenetic response characteristics in the human retina, in keeping with what has been previously observed in ChR2-expressing RGCs in *rd1* mice retinae (Thyagarajan *et al*, 2010). Moreover, the distribution of sustained vs. transient responses of optogenetically activated RGCs could depend on retinal eccentricity. More experiments would be necessary to assess the significance of these observations for the human retina.

We utilized both lentiviral and AAV vectors to target the human retina with ReChR. The reason for using the lentiviral vector was to accelerate the expression of the optogenetic tool under cell culture conditions, so that the light responses of ReChR-expressing cells could be studied after a few days of viral incubation. However, from a translational perspective, we could demonstrate that the clinically feasible AAV mediates expression of functional ReChR in the human retina. We showed that AAV-mediated ReChR in the human retina could drive light responses after 12 days of viral incubation *ex vivo*, using an explant prepared from the peripheral human retina. Studies of intravitreal injections of AAV in macaques have reported that the target gene is expressed mainly within a small circular ring around the fovea center (Yin *et al*, 2011, 2014). The inner limiting membrane (ILM), that covers the surface of the primate retina, is a barrier to AAV spread *in vivo* (Dalkara *et al*, 2009) and limits pan-retinal transduction of AAV in the primate. In future studies, it should be investigated whether this issue could be mitigated by designing AAV vectors optimized for the primate retina. Given that current AAV technology is sufficient to transduce the primate para-fovea *in vivo*, we assume that at least the para-foveal region could be targeted in human subjects with an AAV encoding ReChR, in order to reactivate the high-acuity midget RGC receptive fields in blind patients.

# Materials and Methods

### Calculation of safety thresholds for optogenetic stimulation of the retina

Our calculations of the safety thresholds of spectrally adjusted retinal irradiance regarding potential photochemical damage are based on the 2006 European Directive on artificial optical radiation (European Commission, 2006). This EU-Directive is in agreement with the guidelines of the International Commission on Non-Ionizing Radiation Protection (on limits of exposure to incoherent visible and infrared radiation) (ICNIRP, 2013).

According to the EU-Directive and the ICNIRP guidelines, the maximum safe radiance of an emitter is given by:

$$L_B = \sum_\lambda L_\lambda(\lambda) B_\lambda(\lambda) \leq 10\,\mathrm{mW\,cm^{-2}\,sr^{-1}} \qquad (1)$$

where $L_B$ is the wavelength-corrected equivalent radiance (W cm$^{-2}$ sr$^{-1}$), $L_\lambda$ is the source radiance at a specific wavelength, and $B_\lambda$ is the spectral weighting function. Equation (1) gives the limit for continuous radiation.

It has been shown previously (Sliney *et al*, 1995; Degenaar *et al*, 2009) that the source radiance (W cm$^{-2}$ sr$^{-1}$) can be converted into the irradiance on the retina $E_r$ (W cm$^{-2}$) by the following equation:

$$E_r = \pi L_s \tau d_e^2 / 4f^2 \qquad (2)$$

where $L_s$ is the source radiance (W cm$^{-2}$ sr$^{-1}$), $\tau$ is the transmittance of the ocular media ($\sim 0.9$ in young individuals), $d_e$ is the pupil diameter ($\sim 0.3$ cm), and $f$ is the effective focal length of the eye ($\sim 1.7$ cm). Then, we can assume the following relationship:

$$E_r = 0.02 L_s \qquad (3)$$

In units of W cm$^{-2}$ and from equation (1), we get:

$$E_B = \sum_\lambda E_\lambda(\lambda) B_\lambda(\lambda) \leq 200\,\mu\mathrm{W\,cm^{-2}} \qquad (4)$$

where $E_B$ is the wavelength-corrected equivalent irradiance (W cm$^{-2}$) of the retina. This should not exceed 200 µW cm$^{-2}$.

By using Planck's constant and the spectral weighting factor of photochemical hazard at a particular wavelength, given by the international guidelines (European Commission, 2006; ICNIRP, 2013), we can calculate the safety thresholds for specific wavelengths in photons cm$^{-2}$ s$^{-1}$. Our calculations (see Table 1) show that we are allowed to increase the light intensity for stimulation by three orders of magnitude (European Commission, 2006; ICNIRP, 2013) if we shift the wavelength from blue to orange light.

### Animals

All mice used in this study were C3H/HeN (*rd1* mice) (Janvier Laboratories, Le Genest Saint Isle, France), a genetic mouse model of fast photoreceptor degeneration. Mice were housed in a 14-h light/10-h dark cycle with *ad libitum* access to food and water. Macaque retinal explants were prepared using eyes received from adult macaques (*Macaca fasicularis*) of foreign origin that were terminally anesthetized for unrelated studies. All animal experiments and procedures were approved by the local animal experimentation ethics committee (Le Comité d'Ethique pour l'Expérimentation Animale Charles Darwin) and were carried out according to institutional guidelines in adherence with the National Institutes of Health guide for the care and use of laboratory animals as well as the Directive 2010/63/EU of the European Parliament.

### Plasmids, AAV and lentivirus production

The plasmids encoding a fusion protein of ReChR and the reporter mCitrine under the human Synapsin (hSyn) promoter in an adeno-associated virus (AAV) backbone and as a lentiviral shuttle plasmid, that is, AAV-hSyn:ReChR-mCitrine and pLenti-hSyn:ReChR-mCitrine were provided by Dr. Roger Tsien, University of California, San Diego). For the primate retinal explant experiments, the coding sequence of ReChR was PCR amplified from the above AAV plasmid and subcloned using the In-Fusion method (In-Fusion HD Cloning Kit, Clontech Laboratories, Mountain View, CA) into an existing vector backbone containing the ubiquitous CAG promoter to generate AAV-CAG-ReChR-GFP. The In-Fusion primer sequences

were as follows: forward (5′ to 3′): CGGCCGATCCACCGGTGGTGAC CATGGTGAGCAGA, reverse (5′ to 3′): CATGGTGGCGACCGGTAGG CTGCTCTCGTACTTATCTTC. Recombinant AAVs were produced for both the hSyn and CAG promoter-driven constructs using the plasmid cotransfection method, and the resulting lysates were purified via iodixanol gradient ultracentrifugation as described previously (Mace et al, 2015). AAVs were pseudotyped with AAV2 and AAV8 Y447 733F capsids (Kay et al, 2013) for the hSyn and CAG constructs, respectively. Lentiviral particles for the hSyn:ReaChR-mCitrine construct were produced by transient cotransfection of 293T cells using the ReaChR-expressing lentiviral shuttle plasmid, an HIV-1-derived packaging plasmid and VSV-G-envelope-expressing plasmid as previously described (Bemelmans et al, 2005). Forty-eight hours post-transfection, the lentiviral particles were harvested in the culture medium and were concentrated by ultracentrifugation. Viral titers were estimated using the HIV-1 p24 antigen assay (Zeptometrix, Buffalo, NY) according to the manufacturer's instructions.

### Intravitreal AAV injections in mice

Rd1 mice, 4–5 weeks old, were anesthetized with ketamine (50 mg/kg) and xylazine (10 mg/kg). Intravitreal injection was performed bilaterally: Pupils were dilated, and an ultrafine needle was passed through the sclera, at the equator and next to the limbus, into the vitreous cavity in order to inject 2 μl AAV2-hSyn:ReaChR-mCitrine AAV ($2.53 \times 10^{13}$ vector genomes $ml^{-1}$).

### Primate retinal explant culture and AAV infection

For primate retinae, the eyes of terminally anesthetized macaques were removed and transported in $CO_2$-independent medium (Life Technologies, Carlsbad, CA). Upon arrival, they were dissected under room light and processed for tissue culture (similar to the processing of human retinal explants from Busskamp et al, 2010). The anterior parts were removed, and the vitreous humor with the attached neural retina was transferred to $CO_2$-independent medium. The retina was isolated from the vitreous humor and cut into ~1-$cm^2$ pieces. With the RGC face up, the retinal pieces (from the peripheral retina) were mounted on the polycarbonate membrane of a Transwell (Corning Inc., Corning, NY) 0.4-μm cell culture insert with one drop of medium and flattened with a polished Pasteur pipette. Subsequently, the $CO_2$-independent medium was removed and replaced with 2 ml Neurobasal-A medium supplemented with 2 mM L-glutamine and B-27 (NBA+, Life Technologies) in each well containing mounted tissue. Macaque retinal explants were infected with 10 μl AAV2-hSyn:ReaChR-mCit ($2.53 \times 10^{13}$ vector genomes $ml^{-1}$) (same vector used for in vivo mouse injections) and 10 μl AAV8 Y447 733F-CAG:ReaChR-GFP ($3.1 \times 10^{12}$ vector genomes $ml^{-1}$) added on top of the retinal explant and kept in a tissue culture incubator (37°C, 5% $CO_2$) until the day of electrophysiological recordings or fixation. The AAV infections were performed within 2 h of the retinal explants being put in tissue culture.

### Human retina and lentivirus infection

Postmortem human ocular globes from donors were acquired from the School of Surgery (Ecole de Chirurgie, Assistance Publique Hôpitaux de Paris) via a protocol approved by the institutional review boards of the School of Surgery and the Quinze-Vingts National Ophthalmology Hospital (CPP Ile-de-France V). All experiments on postmortem human retinal explants were performed according to local regulations as well as the guidelines of the Declaration of Helsinki. Eyes were collected from six anonymous donors (ranging from 63 to 95 years of age) at postmortem delays ranging from 9 to 38 h. The eyes were transported in $CO_2$-independent medium and upon arrival were dissected under room light and processed for tissue culture exactly according to the procedure used for the non-human primate retinae. From each donor retina, peripheral pieces were tested on the multi-electrode array to identify viable retinae exhibiting spontaneous RGC activity. The macula from one retina (received at a postmortem delay of 10 h) exhibiting RGC spontaneous spiking was dissected through the foveal pit (see Fig 6A and B), and this piece was infected with the ReaChR lentivirus. For targeting ReaChR to postmortem human RGCs, a lentivirus was chosen because it has been shown this vector can induce fast gene expression in retinal cell culture, even when tissue-specific promoters are utilized (Busskamp et al, 2010). About 10 μl of the hSyn:ReaChR-mCitrine lentivirus ($2.26 \times 10^9$ vector genomes $ml^{-1}$) was added on top of the explant (RGC side up) prepared from the macular region. This construct encoded ReaChR under the pan-neuronal hSyn promoter (same as the one used for in vivo mouse experiments). The viral infection was performed within 24 h of the explant being put in tissue culture.

### Human retina and AAV infection

Explants were prepared from the far periphery of a donor human retina (received ~9 h postmortem) utilizing procedures described in the above section. These pieces were infected with 10 μl AAV2-hSyn:ReaChR-mCit ($2.53 \times 10^{13}$ vector genomes $ml^{-1}$) (same vector used for in vivo mouse injections) and 10 μl AAV8 Y447 733F-CAG:ReaChR-GFP ($6.26 \times 10^{13}$ vector genomes $ml^{-1}$) and kept in tissue culture (37°C, 5% $CO_2$ incubator), similar to the procedure described above until the day of the electrophysiological recordings or fixation.

### Fundus imaging

About 4–5 weeks after intravitreal delivery, in vivo retinal expression of mice injected with AAV2-hSyn:ReaChR-mCitrine was checked using in vivo fundus imaging (MicronIII, Phoenix Instruments, CA). Mice were kept lightly anesthetized under isoflurane.

### Fluorescence and confocal microscopy

Eyecups from ReaChR-treated rd1 mice were harvested at 2–4 months post-injection for immunofluorescence. The eyecups were fixed in 4% paraformaldehyde for 30 minutes at room temperature followed by three washes in PBS at room temperature and stored overnight in PBS containing 30% (w/v) sucrose (Sigma-Aldrich, St. Louis, MO). The eyecups were then dissected in PBS, and the anterior segments of the eye were removed. The neural retina together with the retinal pigment epithelium were embedded in 4% agarose and 30% sucrose (w/v in PBS) and frozen in −80°C until sectioning. The frozen blocks were then sectioned in a cryostat, and 12-μm retinal sections were mounted on slides for confocal

imaging. Prior to imaging, the retinal sections were fixed in 4% PFA for 30 min, rinsed 5× with PBS, incubated with DAPI (1:5,000, Invitrogen, Carlsbad, CA) and washed in PBS three times, and coverslipped with Vectashield mounting medium (Vector Laboratories, Burlingame, CA). Primate retinal explants were also processed as above except that the retinal explant pieces were fixed prior to embedding in the sectioning medium. The human retinal explant was imaged as a whole mount (after DAPI staining) using the same foveal explant piece in which light-evoked responses were observed. The human retinal explant infected with AAV was cut into vertical sections. Images of the 12-μm vertical sections of ReAChR-treated *rd1* retinae were acquired in an inverted laser scanning confocal microscope (FV1000, Olympus, Tokyo, Japan) for which pulsed lasers with wavelengths 405, 488, and 515 nm were used to visualize the DAPI, the endogenous GFP, and mCitrine fluorescences, respectively, of the ReAChR's reporter tag. The images were acquired sequentially line by line, and the step size was optimized based on the Nyquist–Shannon theorem. Images were analyzed in FIJI (NIH, Bethesda, MD), and Z-sections were projected on a 2D plane using the MAX intensity setting in the software's Z-project feature. Low magnification images of the live endogenous fluorescence of the human retinal explant were captured using a Nikon AZ100M Macro fluorescence microscope (Nikon Corp., Tokyo, Japan).

### 2-photon imaging and patch-clamp recordings

A custom-made 2-photon microscope equipped with a 25× water immersion objective (XLPlanN-25x-W-MP/NA1.05, Olympus) and a pulsed femto-second laser (InSight™ DeepSee™, Newport Corporation, Irvine, CA) was used for imaging and patch-clamp recording of mCitrine or GFP-positive ganglion cells. Mice were sacrificed by $CO_2$ inhalation followed by quick cervical dislocation, and eyeballs were removed. ReAChR-treated retinae from *rd1* mice were isolated in oxygenated (95% $O_2$/5% $CO_2$) Ames medium (Sigma-Aldrich) and placed in the recording chamber of the microscope at 36°C for the duration of the experiment. For live 2-photon imaging, a whole-mount retina with ganglion-cell-side up was placed in the recording chamber of the microscope. Image acquisition was made using the excitation laser at a wavelength of 930 nm (Fig 1B). For patch-clamp recordings, a CCD camera (Hamamatsu Corp., Bridgewater, NJ) was used to visualize the retina under infrared light, and AAV-transduced fluorescent ganglion cells were targeted with a patch electrode under visual guidance using the reporter tag's fluorescence. Whole-cell recordings were made using an Axon Multiclamp 700B amplifier (Molecular Device Cellular Neurosciences, Sunnyvale, CA). Patch electrodes were made from borosilicate glass (BF100-50-10, Sutter Instrument, Novato, CA) pulled to 5-10 MΩ, and filled with 112.5 mM $CsMeSO_4$, 1 mM $MgSO_4$, $7.8 \times 10^{-3}$ mM $CaCl_2$, 0.5 mM BAPTA, 10 mM HEPES, 4 mM $ATP-Na_2$, 0.5 mM $GTP-Na_3$, 5 mM lidocaine *N*-ethyl bromide (QX314-Br) (pH 7.2). Excitatory currents were isolated by voltage clamping ganglion cells at the reversal potential of $Cl^-$ (−60 mV). A few ganglion cell spike recordings were performed with a cell-attached configuration, using the same electrodes, but filled with Ringer's medium. For macaque retina, only cell-attached recordings were performed (Fig 5E). The macaque tissue was also superfused in Ames medium bubbled with 95% $O_2$ and 5% $CO_2$ at 34°C for the duration of the experiment.

Metabotropic glutamate receptor agonist L-2-amino-4-phosphonobutyric acid (L-AP4, Tocris Bioscience, Minneapolis, MN) was bath-applied in the perfusion system at a concentration of 50 μM in order to block any potentially remaining synaptic input from the photoreceptors to the ON bipolar cells. Light intensity was modulated by neutral density filters over a range of 6 log units (ND 0-ND 50) using a digital light projector (V-332, PLUS Corp., Beaverton, OR) and a 590BP20-nm filter. A monochromatic light source (Polychrome V, TILL photonics (FEI), Hillsboro, OR) was used to measure the activity spectrum of ReAChR. We used 300-ms light flashes ranging from 650 to 400 nm (25 nm steps; interstimulus interval 1.5 s) at a constant light intensity of $1.2 \times 10^{16}$ photons $cm^{-2}$ $s^{-1}$. The monochromatic light source was also used to generate light pulses at different frequencies in order determine the temporal response properties of ReAChR. Stimuli were generated using custom-written software in LabVIEW (National Instruments, Austin, TX). Output light intensities were calibrated ($10^{12}$–$10^{18}$ photons $cm^{-2}$ $s^{-1}$) by using a spectrophotometer (USB2000+, Ocean Optics, Dunedin, FL).

### Multi-electrode array recordings and data analysis

All multi-electrode array recordings were performed on a 252 channel multi-electrode array system (USB-MEA256-System, Multi Channel Systems, Reutlingen, Germany), and data were acquired using the MC_Rack software (MC_Rack v4.5, Multi Channel Systems). *Ex vivo* isolated flat-mounted retinae of ReAChR-treated *rd1* mice aged 2 months or older as well as ReAChR-treated primate *ex vivo* retinal explants in tissue culture were tested. Mice were euthanized as described above and post-dissection, the retinae were placed on a cellulose membrane soaked overnight in poly-L-lysine (Sigma-Aldrich) and superfused in oxygenated Ames medium (A1420, Sigma-Aldrich) containing sodium bicarbonate (Sigma-Aldrich) at 34°C, with the RGC side gently pressed against a 60-μm electrode spacing multi-electrode array chip (256MEA60/10iR, Multi Channel Systems). Full-field light stimuli were presented using a Polychrome V monochromator (TILL Photonics, Munich, Germany) and driven by a STG2008 stimulus generator (Multichannel Systems), using custom-written stimuli in MC_Stimulus II (MC_Stimulus II version 3.4.4, Multi Channel Systems). Output light intensities were calibrated using a spectrophotometer (USB2000+, Ocean Optics, Dunedin, FL). To construct the action spectrum of ReAChR in mice, the retinae were stimulated with light wavelengths ranging from 400 to 650 nm at 10 nm steps for a duration of 2 s per presentation at ~$10^{16}$ photons $cm^{-2}$ $s^{-1}$ light intensity with an interstimulus interval of 10 s. The order of presentation of individual wavelengths was scrambled to minimize the possibility of adaptations in the retina, and each step was tested 10 times. A flicker stimulus consisting of light at 550-nm blinking at frequencies ranging from 1 to 100 Hz (intensity: $1.59 \times 10^{16}$ photons $cm^{-2}$ $s^{-1}$) was presented to the retinae to measure the temporal dynamics of ReAChR's light response in ReAChR-treated blind mice. RGC activity was then amplified and sampled at 20 kHz. Signals were filtered with a 200 Hz high-pass filter in MC_Rack, and individual channels were spike-sorted by visualizing the waveforms' principal component analysis plots using Spike2 software v.7 (Cambridge Electronic Design Ltd., Cambridge, UK). Custom-written MATLAB scripts were used for plotting the responses to randomized presentations of different wavelengths (10 nm steps) of light and for analyzing the

responses to flicker stimuli. For a given flicker frequency tested, a spike modulation index (M.I.) was calculated based on the ratio of the difference of the maximum ($f_{max}$) and minimum ($f_{min}$) firing rates during the stimulation to the cell's mean spontaneous firing rate ($f_{mean}$), that is, M.I. = $(f_{max} - f_{min})/(f_{mean})$. The M.I. of each cell measured for a range of flicker frequencies (1–100 Hz) was normalized to its maximum M.I. to obtain a cell's modulation ratio. The mean modulation ratio of ReAChR-responding cells is presented ($n = 4$ mice retinae, 297 cells) in Fig 2F; error bars were calculated over cells. For plotting the action spectrum in Fig 2D (dashed line), the firing rate of a light-responding cell during a one-second window after the stimulus, was recorded. The response of each cell was normalized by its maximum firing rate. Error bars were calculated over different animals ($n = 5$ mice).

For the primate experiments (macaque and human), retinal explants infected with AAV or lentivirus in tissue culture for 3–12 days were utilized by which time the innate light responses of the retina are undetectable in the multi-electrode array. For the primate experiments, immediately prior to the experiment, a macaque or human retinal explant piece that had been infected with the CAG:ReAChR AAV or hSyn:ReAChR lentivirus was dismounted from the Transwell membrane (Corning, Tewksbury, MA) and mounted on a cellulose membrane soaked overnight in poly-l-lysine (Sigma-Aldrich) and gently lowered on a 100 μm multi-electrode array (Multichannel systems). Tissues were superfused in oxygenated Ames medium (containing sodium bicarbonate) at 34 °C. Data are presented from recordings obtained after L-AP4 was added to the medium to block any remaining input from the photoreceptors to ON bipolar cells. L-AP4 was freshly diluted to a final concentration of 50 μM and bath-applied through the perfusion system for at least 15 min prior to recordings. The action spectrum of ReAChR was constructed in these retinae similar to the procedure describe for mouse retinae, that is, same stimulus and data analysis methods. The error bars were calculated over three individual macaques. To assess, if these retinae can be safely activated at intensities below the safety threshold, the ReAChR-expressing primate retinae were presented with full-field stimulus (2 s duration, 10 repetitions, interstimulus interval of 10 s) of light at 550 nm (intensities: $1.59 \times 10^{16}$ photons cm$^{-2}$ s$^{-1}$) and 580 nm (intensity: $1.24 \times 10^{17}$ photons cm$^{-2}$ s$^{-1}$). The signal processing for spike detection and sorting was similar to those applied to the multi-electrode array recordings from mice retinae. The raster plots and peristimulus time histogram data (bin size of 100 ms, standard error of mean over channels) presented for the macaque and human data were constructed in MATLAB using custom scripts from spike-sorted channels and further processed in Adobe Illustrator CS5 (Adobe Systems, San Jose, CA) for presentation.

### Visual cortex extracellular recordings

As previously described in detail (Mace *et al*, 2015), mice (3–9 months old) were sedated with ketamine–xylazine injection and deeply anesthetized with urethane. Blocked by a stereotaxic holder, the mice were maintained at 37°C (heating pad controlled by a rectal probe). Both eyes (with pupils dilated) were covered with small coverslips to avoid dehydration. A 1-mm$^2$ craniotomy was performed above V1 in the left hemisphere (3 mm lateral and 0.5 mm rostral from the lambda point). The dura was removed, and

the exposed area was maintained, cleaned, and hydrated. Extracellular recordings were made with a silicon multisite (16) electrode (A1 × 16-3 mm-50-703, NeuroNexus Technologies, Ann Arbor, MI), in order to record simultaneously multiple layers of cortex (spikes and LFP). The multi-electrode was gently inserted into the cortex perpendicularly to the surface of the brain and 600 μm deep. Then, the tissue surface was covered with agarose (1.2% in cortex buffer). Signals were amplified using a 16-channel amplifier from MultiChannel Systems (model ME16-FAI-μPA-System), sampled at 25 kHz, and recorded using the software MC Rack (Multi Channel Systems, Reutlingen, Germany). For spikes and for LFP, a high-pass filter at 200 Hz and a low-pass filter at 300 Hz were used, respectively. Signals were then analyzed using custom MATLAB (Mathworks, Natick, MA) scripts. For every experiment, the electrode presenting the best signal was selected and analyzed. LFPs were shown as an average of 200 repetitions, spiking events detected by a threshold set at six times the median absolute deviation of the trace and peristimulus time histograms generated with bins of 5 ms. For light stimulations, a custom stimulation device was placed 1 cm from the eye of interest with its illumination restricted only to that eye. The stimulation consisted of 200 ms pulses of orange light (595 nm collimated LED, M595L3, Thorlabs, Dachau, Germany) repeated 200 times at 1 Hz and triggered by a Digidata (Axon; Molecular Devices, Sunnyvale, CA). The range of light intensities (modulated with neutral density filters) as measured at the level of the cornea was between ($2.5 \times 10^{13}$ and $2.5 \times 10^{17}$) photons cm$^{-2}$ s$^{-1}$.

### Light-induced behavior

A circular open-field arena was constructed in house with diameter 36 cm using an aluminum sheet folded in a circle placed over a plexiglass base (similar to the one described in Polosukhina *et al*, 2012). Ten 590-nm LED light sources (Cree XP-E, Lumitronix, Graefelfing, Germany) were glued to the wall of the open-field arena. The irradiance of the open field upon LED illumination was ~$10^{15}$ photons cm$^{-2}$ s$^{-1}$ as measured approximately at the level of the mice's eyes using a handheld spectrophotometer (PM100D, Thorlabs, Germany). Activity in the arena was recorded during the test by a video camera (Handycam, Sony Corporation, Tokyo, Japan) in night vision mode. ReAChR-treated *rd1* mice or age-matched blind *rd1* control mice were dark adapted for 30 min before being placed in the center of the open-field arena. Their activity was recorded for 2 min followed by 2 min of illumination of the chamber (based on the protocol used in Lagali *et al*, 2008). The movies were analyzed using mouse behavior phenotyping and motion tracking software Ethovision XT build 8.0 (Noldus Information Technology, Leesburg, VA). The mouse was detected using a center of mass algorithm by adjusting the threshold to maximize the contrast between the mouse and background, in the dark and during illumination. The average velocity and trajectory of the mouse was tracked for 2 min in the dark and for 2 min after light was switched on. All behavior experiments were run during the dark phase of the mice between 18 and 21 h. These data were then analyzed and plotted in Prism 6 (GraphPad Software, La Jolla, CA). Significance was assessed using a one-tailed Student's *t*-test, and an α-level of $P < 0.05$ was considered to be significant. Cumulative trajectories over 2 min of *rd1* and ReAChR mice moving in the open-field arena in the light onset were plotted using Ethovision to visualize the differences in locomotory

## The paper explained

### Problem

Channelrhodopsin, an algae-derived blue light-gated ion channel, holds promise for treating blindness caused by retinal degenerative diseases, such as retinitis pigmentosa. Since its discovery, channelrhodopsin-2 has been adopted in neuroscience research as a crucial tool for probing neural circuits. Significant advances have also been made to improve its membrane targeting and response kinetics that have enhanced its potential as a research tool. However, the high intensity of blue light required to elicit responses may cause photochemical damage in the retina and breaches the safety threshold stipulated by regulatory agencies. This might impose a significant practical limit to its advancement to the clinic for vision restoration. Importantly, the damage potential of red-shifted light is vastly lower than that of blue light. Recently, a red-shifted ChR2 has been developed, which we tested for a safe therapeutic approach to recover vision.

### Results

Here, we show orange light-evoked responses in a mouse model of retinitis pigmentosa expressing a novel red-shifted channelrhodopsin, ReaChR, in retinal ganglion cells. This light stimulation, while being under the safety threshold, triggers neural responses in the mouse visual cortex and induces light avoidance behavior. We found that this red-shifted channelrhodopsin also drives neuronal responses in macaque retinae as well as in the central human retina, the site of high-acuity vision, demonstrating the therapeutic potential of the red-shifted channelrhodopsin molecule.

### Impact

We showed in our translational approach, that red-shifted ChR2s are capable of triggering retinal light responses under safe illumination intensities, not only in mouse, but also in macaque and human *ex vivo* retinae.

behavior between ReaChR-treated and blind *rd1* mice upon light onset. The group averages of mean velocities of mice (*rd1*, $n = 12$; ReaChR-treated, $n = 11$) were compared using one-tailed *t*-tests. For the light/dark box experiments, a custom-made light/dark box was constructed (similar to Bourin and Hascoët (2003) but adapted to perform optogenetic stimulation at 590 nm). A box of dimensions, 36 cm (l) × 20 cm (b) × 18 cm (h), was divided lengthwise into two equal sized compartments using a non-transparent wall with a 7 cm (b) × 5 cm (h) hole in the middle. The light compartment was fitted with eight 590-nm LEDs (Cree XP-E, amber, Lumitronix), on an aluminum heat sink, at a height of 3 cm from the floor of the cage. All mice were age-matched, and their ages ranged between 7–8 weeks at the time of testing. Male *rd1* ($n = 9$), male WT C57BL6J ($n = 9$), and male ReaChR-treated ($n = 7$) mice were subjected to 5-min trials during which the compartment was illuminated at different light intensities of orange (590 nm) light. The mice were habituated to the testing room in dark for 2 h prior to testing, and all tests were performed during the dark phase (19–0 h). The behavior of mice were tested for a range of light intensities from $2.07 \times 10^{13}$ to $2.05 \times 10^{16}$ photons cm$^2$ s$^{-1}$ at approximately log unit steps. Light intensities in the light compartment were adjusted using an adjustable voltage supply (VLP-1303 PRO, Voltcraft) and were measured using a spectrophotometer (Ocean Optics). Light intensity measurements are averaged from three locations in the light compartment, 1 cm from the LED, at the

center of the cage and immediately next to the hole on the illuminated side. Observers blind to the experimental procedures and treatment groups analyzed the behavior of the mice manually. Mice were introduced individually in the light compartment and allowed to freely explore the box. The position of the mouse's head was used to define the compartment that it occupied. The time spent in each compartment was recorded. If an animal never crossed the barrier in the first 3 min in the dark, it was excluded from the trial. Statistical significance was assessed by an ordinary one-way ANOVA, the Tukey's multiple comparison test was used to compare group means, and α-level of $P < 0.05$ was considered significant.

**Expanded View** for this article is available online.

## Acknowledgements

We thank Roger Y. Tsien for providing the ReaChR plasmids. We thank Benjamin Taklifi and Céline Winckler for technical support and Elric Esposito and Stéphane Deny for helpful discussions. We are thankful to Noga Vardi for critical feedback. We thank Manuel Simonutti in the Plateforme Animalerie at the Institut de la Vision for performing intravitreal AAV injections. We are grateful to Kate Grieve for organizing the human donor eyes. This study was supported by the Centre National de la Recherche Scientifique (CNRS), the Institut National de la Santé et de la Recherche Médicale (INSERM), Pierre et Marie Curie University (UPMC), an Agence Nationale de la Recherche Grant (LIFESENSES: ANR-10-LABX-65) and an ERC Starting Grant (OPTOGENRET, Grant No 309776).

## Author contributions

AS designed experiments; performed majority of the experiments, including, all human electrophysiological recordings and macaque multielectrode recordings; procured and dissected human donor eyes; generated lentiviral particles; performed AAV injections, *ex vivo* AAV and lentiviral transfections of macaque and human retinae, fundus imaging, histology and confocal microscopy; built behavior setups and performed all behavior assays; developed MATLAB scripts; analyzed data; and wrote the paper. AC performed patch-clamp recordings in mice and macaque, 2-photon imaging in mice and analyzed data. EM performed *in vivo* cortical recordings and analyzed data. RC performed multielectrode recordings in mice. MD generated AAV particles. ML performed *in vivo* injections in mice and confocal microscopy. VF and SP provided macaque retinal explants. OM developed MATLAB scripts and analyzed data. JYL developed ReaChR. J-AS provided macaque retinae. DD designed experiments and generated AAV particles. JD conceptualized the study, designed experiments, and wrote the paper.

## Conflict of interest

DD is a consultant for GenSight Biologics. SP and JAS are founders of GenSight Biologics.

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
