## [Review Process File · EMBO Molecular Medicine]

Red-shifted channelrhodopsin stimulation restores light responses in blind mice, macaque retina, and human retina

Abhishek Sengupta, Antoine Chaffiol, Emilie Macé, Romain Caplette, Mélissa Desrosiers, Maruša Lampič, Valérie Forster, Olivier Marre, John Y. Lin, José-Alain Sahel, Serge Picaud, Deniz Dalkara, and Jens Duebel

Corresponding author: Jens Duebel and Deniz Dalkara, Institut de la Vision

Review timeline:

Submission date:	03 August 2015
Editorial Decision:	09 September 2015
Revision received:	14 January 2016
Editorial Decision:	12 February 2016
Revision received:	14 June 2016
Editorial Decision:	18 July 2016
Revision received:	09 August 2016
Accepted:	11 August 2016

Transaction Report:

Editor: Roberto Buccione

1st Editorial Decision

09 September 2015

Thank you for the submission of your manuscript to EMBO Molecular Medicine. We have now heard back from the three Reviewers whom we asked to evaluate your manuscript. We are sorry that it has taken longer than usual to get back to you on your manuscript. In this case we experienced some difficulties in obtaining the Reviewer evaluations in a timely manner. Further to this, I wished to discuss the evaluations with my colleagues.

As you will some of the issues raised are fundamental and in part shared by the three Reviewers. Although I will not dwell into much detail, I would like to highlight the main points.

Reviewer 1 would like to see more documentation of the data from wild type mice to better compare to the treated, including the behavioral assays. In this respect, the Reviewer also suggests additional tests (see also my comments on reviewer 2's evaluation). The reviewer also suggests additional improvements for you to act upon.

Reviewer 2 is much more reserved and raises a number of concerns including significant dissatisfaction on the overall clinical relevance of your data, and the novelty of the message. S/he mentions that bipolar cells, rather than RGC would be the target of choice. I do not necessarily endorse this view, but I agree that there have been studies supporting this and I therefore suggest

that you should fully explain the rationale for focusing on the RGC rather than bipolar cells. Furthermore (and as mentioned also by Reviewer 1), I agree that it would be important to better characterize the visual discriminations with a cognitive component. I should also mention that many of Reviewer 2's points concerning the presentation of results and discussion are very well taken. There is indeed much margin for improvement in terms of readability, the toning down of over-reaching statements and also improving access for non-experts. That said, I would encourage you to address all comments carefully, although additional experimentation on the other issues may be considered further-reaching.

Reviewer 3, lists several items that require your action and should prove useful to increase the solidity of the conclusions. I note that, again, dissatisfaction is expressed with the behavioral assay and also that the rationale for focusing on the RGC rather than bipolar cells should be provided.

In conclusion, while publication of the paper cannot be considered at this stage, given the potential interest of your findings, we have decided to give you the opportunity to address the above concerns.

We are thus prepared to consider a substantially revised submission, with the understanding that the Reviewers' concerns must be addressed with additional experimental data where appropriate and that acceptance of the manuscript will entail a second round of review. The overall aim is to significantly upgrade the impact, significance and translational relevance of the dataset, which of course are of paramount importance for our title.

I understand that if you do not have the required data available at least in part, to address the above, this might entail a significant amount of time, additional work and experimentation and might be technically challenging, I would therefore understand if you chose to rather seek publication elsewhere at this stage. Should you do so, we would welcome a message to this effect.

Please note that it is EMBO Molecular Medicine policy to allow a single round of revision only and that, therefore, acceptance or rejection of the manuscript will depend on the completeness of your responses included in the next, final version of the manuscript.

As you know, EMBO Molecular Medicine has a "scooping protection" policy, whereby similar findings that are published by others during review or revision are not a criterion for rejection. However, I do ask you to get in touch with us after three months if you have not completed your revision, to update us on the status. Please also contact us as soon as possible if similar work is published elsewhere.

Please note that EMBO Molecular Medicine now requires a complete author checklist (<http://embomolmed.embopress.org/authorguide#editorial3>) to be submitted with all revised manuscripts. Provision of the author checklist is mandatory at revision stage; The checklist is designed to enhance and standardize reporting of key information in research papers and to support reanalysis and repetition of experiments by the community. The list covers key information for figure panels and captions and focuses on statistics, the reporting of reagents, animal models and human subject-derived data, as well as guidance to optimise data accessibility.

I look forward to seeing a revised form of your manuscript as soon as possible.

REFEREE REPORTS

Referee #1 (Remarks):

Impact/Innovativeness: This paper confirms previous findings (from Bi et al., for example) that Chr2 can successfully be used in rd1 mice to restore light responses. However, they are the first to demonstrate that optogenetics can also be used in (post mortem) human eyes to trigger spike responses.

Strengths: The researchers tested the efficacy in multiple models, from mouse to macaque to human, which validates their results. Their research directly pertains to patients with end-stage RP and allied

disorders and could influence future treatment strategies.

Weaknesses:

--Nearly all of the figures show data from treated mice. It would be easier to draw conclusions from the figures if the researchers supplemented them with data from wild type mice as well in order to make a comparison.

--Along those same lines, the researchers should include wild type mice in behavioral assays, perhaps using a light/dark box test (Bourin and Hascoet, 2003) with the treated mice and some wild type mice as another experiment to examine the extent of restoration of visual signals.

--In the test corresponding to Fig4, the researchers should perform the experiment again at varying wavelengths and compare the results with the mice's behavior at 590nm to demonstrate the specificity of ChR activity.

Questions: Fig1A: Why is ChR2 expressed to such a greater extent in the dendrites than in the cell membranes? Why does the same staining technique appear so different in the macaque, with hardly any mCit appearing in the IPL (Fig5A)?

Minor Corrections:

--Please explicitly state that MEA stands for multi-electrode array

--Typo: "To demonstrate the clinical translatability of or approach we tested if ReaChR can be functional, when expressed in RGCs of the primate retina."

--Fig1: it seems that B and C are mislabeled and should be switched.

Referee #2 (Comments on Novelty/Model System):

I am afraid that this work neither presents a novel molecular tool nor any progress to advance a potential optogenetic therapy for blindness.

Here my detailed arguments:

(a) ReaChR is an optogenetic tool that has been characterized in detail previously in vitro and in vivo (incl. spectral sensitivity and kinetics; Lin et al., *Nat Neurosci*, 2013)

(b) Optogenetic vision recovery is a blooming field and its feasibility has been shown multiple times in the blind rd1 mouse model (Tyagarajan et al, *J Neurosci*, 2010; Lagali et al, *Nat Neurosci* 2008; Cronin et al., *EMBO Mol Med* 2015, MacÈ et al. *Science Transl Med*, 2015, Gaub et al, *Mol Ther* 2015, etc.)

(c) The state-of-the-art target cells for vision recovery in advanced stages of photoreceptor degeneration are not the RGCs that were chosen as targets in this study, but the retinal bipolar cells, the interneurons between photoreceptors and RGCs. Reason for this is that the intrinsic retinal signal computation can be mostly retained if targeting the Bipolar cells, in particular the important segregation of ON- and OFF pathways. This has been shown and argued for example in MacÈ et al, *Mol Ther* 2015, Gaub et al. *Mol Ther* 2015 and van Wyk et al *PLoS Biol* 2015. In contrary, when RGCs are targeted, it was shown that all RGCs turn into "ON-cells" (by the way, this important point has been completely neglected in this study), which was argued to be problematic (Tyagarajan et al, *J Neurosci*, 2010; Lin et al, *PNAS* 2008). As the genetic tools for bipolar cell targeting (promoters, AAVs) are available, the taken approach by Sengupte et al. appears as a step backward in development.

(d) Although the main message of this manuscript is that ReaChR can be activated by "safe" light intensities, ReaChR is by far not the first available optogenetic tool with which this is possible. In fact, the trend is to move away from channelrhodopsin variants towards vertebrate opsins, which are naturally 2-3 log units more light sensitive than channelrhodopsin, as they are G-protein coupled receptors and not simple ion channels. An additional advantage of mammalian opsins as potential therapeutic tools is the reduction of the possibility of an adverse immune response on long-term expression; this risk is a significant factor when contemplating the use of microbial channelrhodopsins in humans (and again, this important translational point has been completely neglected in this study). Multiple opsin-based optogenetic tools have been shown to be at least equally if not more light sensitive than ReaChR, e.g. melanopsin in RGCs (Lin et al, *PNAS* 2008), Rhodopsin in BPCs (Gaub et al. *Mol Ther* 2015), Opto-mGluR6 in BPCs (van Wyk et al. *PLoS Biol* 2015). To exemplify this, melanopsin was shown to be activated at indoor lighting intensities and Opto-mGluR6 was shown to be fully activated at ambient daylight. Both would be much better suited for patient treatment than ReaChR, which would require, as stated by the authors in the

discussion, the "patient to wear stimulation glasses in order to stimulate with appropriate light intensity".

(e) Gene therapy in the used model systems (rd1 mouse eye, macaque and human retina) has already been applied, even in the in vivo macaque eye in contrast to macaque retinal explants targeted in this study. Relating to the gene therapy approach, I came across some methodological contradictions of the authors themselves compared to their previous publications, for example:

- why did the authors in this study use lentiviruses and not AAVs to infect human retinas? AAVs are clinically favored vectors and the authors showed previously that they can successfully AAV-infect and maintain human retinal explants, which they are unable to do in this study for unknown reasons (Fradot et al., Hum Gene Ther. 2011)?

- why did the authors use an AAV2/8(Y447,733F) capsid variant when they had recently shown that the Y-F capsid mutant AAVs are less efficient in retinal transduction compared to their own AAV2/2(7m8) capsid variant that was shown to transduce the macaque retina very well (Dalkara et al., Science Transl Med 2013)?

(f) The behavioral paradigm used (open field test) is weak and does not comply with state-of-the-art. The open-field test only shows that the mice react to light (a behavior known to be mediated by intrinsically photosensitive RGCs alone), but it does not define the limits of the visual abilities provided by the optogenetic therapeutic. A water maze, for example that was used in many of the above-mentioned studies, would have been a suited experiment to see whether these mice can carry out visual discriminations with a cognitive component. In particular, behavior should have been tested for light sensitivity as it was done for example in van Wyk et al., PLoS Biol 2015.

(g) Milestone references in the field are not indicated - this puts a misleading light on the novelty of this work, particularly as the discussion is massively overstated.

(h) The C3H/HeN rd1 mouse line is, as commonly known, not an optimal mouse model for vision recovery studies, as it carries the nob5 (GRP179) mutation that renders ON-Bipolar cells non-functional. C3H/HeOu for example, an alternative rd1 mouse model, does not carry the nob mutation.

(i) the Methods part "calculation of safety threshold for optogenetic stimulation" is copied 1:1 from "Degenaar et al., J Neural Eng, 2009" but does not apply to the experimental settings of this study! The authors work on retinal explants and not on human patients, but the formulas all include light attenuation factors through the optic apparatus. Therefore, the "safe" light intensity on a directly illuminated retinal explants would be at least 3 log units lower than claimed by the authors.

(j) Many of the chosen methods are "outdated" and don't give novel insights to move the field forward (see also Reply to Authors for details).

(k) The optogenetic vision recovery field is progressing rapidly and translationally oriented studies have to compare different optogenetic approaches under the same conditions, i.e. investigate the validity of viral vectors (e.g. capsid engineered AAVs) & specific promoters from the mouse in macaque and human tissue, compare and quantify expression levels and biodistribution of the optogenetic therapeutic and determine the potential quality of recovered vision. As none of these issues have been addressed in the present study, I do not recommend this submission for publication in the highly ranked journal EMBO Mol Med.

Referee #2 (Remarks):

This manuscript by Abhishek Sengupta et al. describes the use of the red-shifted Channelrhodopsin, ReaChR, to recover light-sensitivity in the retinas of blind mice as well as in macaque and human retinal explants. For this, ReaChR has been introduced into the retina with AAV and lentiviruses under ubiquitous promoters, nonetheless, electrophysiological characterization has been restricted to the retinal ganglion cells (RGCs).

While I appreciate the use of 3 model systems, the mouse, macaque and human, I am somehow disappointed that the potential of these systems to test for translatability was not exploited, e.g. the authors did not test for the specificity of available promoters, the efficiency and biodistribution of AAV transduction, the time to transgene expression, determination of the quality of the potentially recovered vision, etc. In actual fact, the choice of experiments and the manuscript text appear somehow elusive and the data analysis and discussion somehow superficial. I am afraid that the line of argumentation is most surely not understood by a non-expert, as the reasoning is incomplete.
Major Points:

1. The main argument of this study is the "safe" light-activation of ReaChR. ReaChR is, however, by far not the first available optogenetic tool to function in the "safe light intensity" range. In fact, the trend is to move away from ChR2 variants towards vertebrate opsins, which are naturally 2-3 log

units more light sensitive, as they are G-protein coupled receptors and not simple ion channels. An additional advantage of mammalian opsins as potential therapeutics is the reduction of the possibility of an adverse immune response on long-term expression; this risk is a significant factor when contemplating the use of microbial channelrhodopsins in humans. Multiple opsin-based optogenetic tools have been shown to be at least equally if not more light sensitive than ReaChR, e.g. melanopsin in RGCs (Lin et al, PNAS 2008), Rhodopsin in BPCs (Gaub et al. Mol Ther 2015), Opto-mGluR6 in BPCs (van Wyk et al. PLoS Biol 2015). To exemplify this, melanopsin was shown to be activated at indoor lighting intensities and Opto-mGluR6 was shown to be fully activated at ambient daylight. Both would be suited for patient treatment. In contrary, ReaChR, as stated by the authors in the discussion, would require the "patient to wear stimulation glasses in order to stimulate with appropriate light intensity". Above references have to be added to the manuscript and discussed in comparison to ReaChR. Also, the argument of the novelty of "safe light activation" has to be tuned down throughout the manuscript.

2. On the same note: I actually doubt that ReaChR allows for potential treatment with "safe" light intensities, and this is why: The EU Directive was developed for human individuals being exposed to radiation. Formula (2) on p7 includes light attenuation through the optic apparatus, which Degenaar et al., J Neural Eng, 2009 included in their analysis. The authors here, in contrary, work with retinal explants. Therefore, the "safe light intensity threshold" for retinal explants would be shifted to approximately 3 log units lower light intensities. In other words, ReaChR is not "safe". As there seems to be some confusion about light intensities, I would like the authors to include actual values into the formulas for their experimental settings and relate these to the results.

3. As argued by several recent papers (MacÈ et al, Mol Ther 2015, Gaub et al. Mol Ther 2015 and van Wyk et al PLoS Biol 2015), the state-of-the-art target cells for optogenetic vision recovery in advanced stages of photoreceptor degeneration are not the RGCs, but the ON-bipolar cells. Reason for this is that the intrinsic retinal signal computation can be greatly retained when the ON-bipolar cells are targeted, in particular the important segregation of ON- and OFF pathways. In contrary, when RGCs are targeted, it was shown that they all turn into "ON-cells" and that the "transientness" of their response variety is reduced (Tyagarajan et al, J Neurosci, 2010; Lin et al, PNAS 2008). Both of these points, despite their importance for the quality of recovered vision, have been entirely neglected in this study - what percentage of RGCs expressed ReaChR? Did other cells also express (due to the ubiquitous promoters used)? What was the RGC response variety in treated rd1 compared to healthy retinas (ON-, OFF-, transientness)? What are the potential consequences on recovered vision if for example only sustained ON-responses are being recovered? As the genetic tools for bipolar cell targeting (promoters, AAVs) are available, I do not understand why the authors targeted the RGCs in this study. All of the above has to be shown, analyzed and discussed in the manuscript.

4. The C3H/HeN rd1 mouse model is not well suited for vision recovery studies as it is known to possess the nob5 (GRP179, no b-wave) mutation, which renders the ON-Bipolar cell pathway non-functional (Nishiguchi KM et al., Nat Communications, 2015). Vision rescue is therefore only achievable through the OFF-pathway. Is this why the RGCs and not the Bipolar cells have been targeted by the authors? If yes, this has to be mentioned in the text.

5. Gene therapy: I came across some methodological contradictions of the authors themselves compared to their previous publications, for example:

- why did the authors use lentiviruses and not AAVs to infect human retinas? AAVs are clinically favored vectors and the authors showed previously that they can successfully AAV-infect and maintain human retinal explants for sufficient time to expression (Fradot et al., Hum Gene Ther. 2011)?
- why did the authors use an AAV2/8(Y447,733F) capsid variant when they had recently shown that Y-F capsid mutant AAVs are less efficient in retinal transduction compared to their own AAV2/2(7m8) capsid variant that was able to transduce the macaque retina very well (Dalkara et al., Science Transl Med 2013)? These points have to be discussed; as the AAV2/2(7m8) variant emerged from the author's labs, I would like to see at least an AAV2/8(Y447,733F) and an AAV2/2(7m8) transduced macaque (and human) retina side by side. I would also like to have a clear explanation on how long the human retinas could be maximally cultured and why the authors did not achieve the results they described earlier in Fradot et al (2011).

6. Experiments on Primate Retina: I have a few comments and the methodology leaves many open questions:

- Ref 22 (Fradot et al) explains the methodology for the human retinal explants, but a much better Ref, including AAV transduction, parafoveal expression and activity patterns in midgen RGCs

would be "Yin L et al., J Neurosci 34, 2014"

- it is not explained why the CAG promoter instead of the hSyn promoter was used; maybe because Yin et al. (2014), Dalkara et al (2013) and others used the CMV and CAG promoters to transduce RGCs? This should be reasoned and either identical promoters should be used in all three models or the different promoters should be compared in their efficiency and distribution of expression.

- I do not believe that the authors saw expression already 3d after AAV transduction; typically AAV2/2 takes 3-4 weeks to express, AAV2/8 might be slightly faster (Natkunarajah M et al., Gene Ther, 2008. Please adapt the text accordingly.

7. In my opinion, the chosen behavioral paradigm, the open field test, does not suffice as a stand-alone behavioral test. Whilst the open-field test is an established test for photophobia and shows if the mice react to light (a behavior that is known to be mediated by ipRGCs alone, see e.g. Semo M et al., PLoS ONE 2010), the test does not show any visually-guided behavior and does not define the limits of the visual abilities provided by the optogenetic therapeutic. A water maze, for example, as used in many previous studies, would have been a suited experiment to see whether these mice can carry out visual discriminations with a cognitive component, for example distinguish between distinct light patterns (moving versus static patterns, etc.), or simply what acuity the recovered vision has (OKR). Besides additional behavioral paradigms, I would also like to see the "light sensitivity of the behavioral response", as it was shown in van Wyk et al., PLoS Biol 2015, as this is the "real" readout for sensitivity.

8. I am missing a figure and text describing the variety of RGC output, e.g. a statement about the transientness of RGC responses, if all RGCs were turned into ON cells by ReaChR expression (as expected) and what consequences this may have for the potentially recovered vision. Also, I would like to know how many RGCs expressed ReaChR, how the exact biodistribution of expression looked like was and how this influenced the locally measured RGC signals. I would also like to see these points discussed and compared between the 3 model systems (mouse, macaque, human).

9. Experiments on human retina: Why is expression only seen in mid-ganglion RGCs and not also in peripheral RGCs? This makes no sense as the lentivirus was applied panretinally. Additionally, the fovea/parafoveal region may present a "problem zone" for optogenetic vision recovery, as the RGCs are skewed to the side, away from the fovea towards the parafovea; you should hypothesize and discuss what this may mean in terms of recovered vision and a potential therapy-skewing of the picture that requires correction optics?

Minor Points:

1. Although I appreciate that the authors went through the trouble measuring the action spectrum of ReaChR in each setting, I do not understand what their motivation was to do so. The ReaChR action spectrum has been determined in the original publication by Lin et al. (Nat Neurosci 2013) and would not change if nothing is engineered on the protein. Fig 4E for example would totally suffice.

2. As ubiquitous promoters have been used, it is expected that ReaChR is expressed in other cell types than RGCs. Have the authors observed an "expression leak"? This point has been entirely hushed up in this study and must be investigated, stated and discussed.

3. p4, 3rd paragraph: it should be mentioned why input from ipRGCs is not of concern (Semo M et al., PLoS ONE 2010).

4. p.4, 4th paragraph: it must be explained why the promoter was changed from hSyn to CAG - is there any indication that CAG is better or hSyn does not function in the primate?

5. p4, 4th paragraph: is this a spelling error or was expression really observed after 3 days? While I would believe this is true for lentiviral expression, AAVs take in the best case a minimum of 2 weeks to express to my knowledge.

6. How do you explain the "uneven" and restricted ReaChR expression pattern in the retinas - centered expression in the mouse retina (e.g. Fig. 1A) and the parafovea-restricted expression in the human explants (e.g. expanded view figure 4)?

7. The labeling of the "expanded view" figures appears to be wrong? For example, expanded view figure 4 corresponds to figure 6...

8. Human retina: (1) why was only 1 retina further investigated when 8 retinas (from 4 donors) were available? I would have liked to see if the observations from the 1 retina can be generalized. (2) How do the authors explain expression to be restricted to mid-ganglion RGCs of the parafovea? (3) Why did the authors use lentiviruses for transduction, as they had previously shown that they can use AAVs (Fradot et al.) which is in light of translatability a much better suited vector? (4) The text on

p.5, para 2 is not well written, the procedure and argumentation for all the taken restrictions is unclear. (5) Contradiction: "identify retinas exhibiting persisting RGC spiking activity" "Note that the human retina was devoid of innate light responses at the time of isolation" (6) As the RGCs are skewed in the parafoveal region (as there is no RGCs directly below the fovea), what do you think will be the consequences on the quality of recovered vision for a ReaChR-treated patient? Would this skewing have to be corrected for optically?

9. The discussion overstates at multiple places, please lessen your arguments. For example: (1) "excellent vision restoration tool" - ReaChR was only shown to recover light responses, but not "vision", (2) "temporally sufficient retinal responses" - for what sufficient? The open-field test does not require great temporal or spatial resolution. Such a strong argument should be shown behaviorally. (3) "compelling responses at the behavioral level" - actually, only a very basic behavioral paradigm has been used that does not show "behavioral vision" (4) "expression without cytosolic aggregation" - from what you show I see many intracellular aggregations and surprisingly weak labeling of the cytoplasm around the soma. Do you have photomicrographs that agree better with your statement? (5) "we could demonstrate clinical translatability" - I disagree, to foster translatability you should have analyzed expression efficiency, biodistribution, cell specificity, promoter, viral validity in the models in parallel, quality of recovered vision, etc. (6) "the treatment we propose has a high potential to be rapidly translated as a therapeutic approach for clinical trials" - I again disagree, there exist more promising optogenetic approaches and nothing has been shown in this study that moves translatability forward. (7) "first proof of principle that with an optogenetic approach it should be possible to stimulate midget RGCs" - please tone this down, as it was shown previously that AAVs transduce midget RGCs in primates with an optogenetic tool (GCaMP5 ; Yin et al, 2011 and 2014). A bigger concern to me is the fact that you seem to only have transduced this RGC cell type, although you applied the lentivirus panretinally.

Referee #3 (Comments on Novelty/Model System):

This is state-of-the-art research in use of channel opsins to restore vision-like behavior

Referee #3 (Remarks):

This is a most interesting manuscript describing the use of a red-shifted channel opsin for restoring light responses to blind retinas in mice and to pharmacologically blinded retinas in primates including humans. This work is important as the use of a red-shifted channel opsin allows higher light intensities to be used safely for activation; this overcomes the challenge of the original (blue) channel opsins which could only be activated at potentially toxic levels of light.

Major comments:

These experiments are overall well designed and executed.

The behavioral assay is relatively crude and measures sensing of light rather than vision per se. The authors should acknowledge this limitation. Did they perform any two choice forced tests or other more visual tests on these animals?

The paper would be improved by a figure panel for both macaque and human retinas showing MEA recording of non-transfected retinas under L-AP4 blockade to demonstrate complete absence of firing.

The difference between MEA and patch clamp 'action spectra' is interesting. To what do the author attribute this? Could this be due to ipRGCs detected on MEA skewing the spectrum to the blue?

To that point, what the authors present as action spectra appear to be single-fluence relative activity spectra. A true action spectrum requires 1.) performing a complete irradiance response relationship for each wavelength tested, 2.) demonstrating that these are all fit by the same curve (Hill coefficient or Michaelis-Menten curve) and then 3.) plotting the I50 point for each wavelength. Only if all curves have the same form does the principle of univariance hold. The authors should clarify that their curves are 'single fluence relative action spectra' if this is indeed the case.

Finally, several recent studies have reported remarkable vision restoration using AAV-delivered

rhodopsin to ON-bipolar cells (Gaub et al., Mol Ther. 2015 Jul 3. doi: 10.1038/mt.2015.121; Cehajic-Kapetanovic et al., Curr Biol. 2015 Aug 17;25(16):2111-22. doi: 10.1016/j.cub.2015.07.029). It appears the more 'upstream' expression of the opsin restores more vision-like firing behavior to RGCs. The authors may wish to discuss whether their red shifted channel opsin could be targeted in this way to bipolar cells.

Minor suggestions:

- Were animals injected bilaterally or unilaterally?
- In figure EV1 it is mentioned that cells are visualized through the linked mCitrine fluorescent protein, but in panel B it is noted that a GFP-positive cell is being targeted. To be clear, is this an mCitrine-positive cell? Or is GFP used in some cases?
- A reference should be provided for the retinal tropism of AAV8 Y447 733F
- In the legend of Figure 1 it appears panels B and C are switched

1st Revision - authors' response

14 January 2016

Referee #1 (Remarks):

Impact/Innovativeness: This paper confirms previous findings (from Bi et al., for example) that Chr2 can successfully be used in rd1 mice to restore light responses. However, they are the first to demonstrate that optogenetics can also be used in (post mortem) human eyes to trigger spike responses.

Strengths: The researchers tested the efficacy in multiple models, from mouse to macaque to human, which validates their results. Their research directly pertains to patients with end-stage RP and allied disorders and could influence future treatment strategies.

Weaknesses:

--Nearly all of the figures show data from treated mice. It would be easier to draw conclusions from the figures if the researchers supplemented them with data from wild type mice as well in order to make a comparison.

We thank the reviewer for his positive comments. Indeed, we agree that wild-type animals are important for the appreciation of our results. Now, we supplemented the figures with data from wild-type mice on the retinal, cortical and the behavioral levels. Please see updated Figure 2B (retina), updated Figure 3 A, B, C (visual cortex) and updated Figure 4 C, D (behavior).

--Along those same lines, the researchers should include wild type mice in behavioral assays, perhaps using a light-dark box test (Bourin and Hascoet, 2003) with the treated mice and some wild type mice as another experiment to examine the extent of restoration of visual signals.

As suggested by the reviewer, we have performed the light-dark box test, where we included wild type mice as controls (Fig. 4C, D). We performed the light-dark box test using a slightly modified version of the protocol used by Bourin and Hascoet, 2003. With this experiment we could demonstrate that ReaChR treated rd1 mice exhibit light avoidance and are capable of distinguishing light from dark (Fig 4C and D). Moreover, in this experiment we have tested the light sensitivity of the ReaChR treated rd1 mice in comparison to untreated rd1 mice and wild type mice (Fig. 4C).

--In the test corresponding to Fig4, the researchers should perform the experiment again at varying wavelengths and compare the results with the mice's behavior at 590nm to demonstrate the specificity of ChR activity.

We thank the reviewer for this suggestion. We ran more behavioral experiments at varying light intensities in order to determine the light intensity requirements at the behavioral level, rather than performing experiments at varying wavelengths. With our electrophysiological from retinal explants we already determined ReaChR's spectral specificity in treated rd1 mice, macaque and human retina. This data showed that ReaChR's peak activity is at ~550 nm in all species and its activity persists considerably at 600 nm. This led us to select 590 to 595 nm as the optimal wavelength for

doing cortical and behavioral experiments. We estimated that at this wavelength range ReaChR should elicit robust activity while staying well below the photochemical damage light intensity threshold.

Another reason for selecting 590 nm was to avoid the stimulation of the melanopsin intrinsically light sensitive ganglion cell population, which would be the only potential remaining source of light sensitivity in the rd1 mice.

Questions: Fig1A: Why is ChR2 expressed to such a greater extent in the dendrites than in the cell membranes? Why does the same staining technique appear so different in the macaque, with hardly any mCit appearing in the IPL (Fig5A)?

By using live two-photon imaging, we observed ReaChR expression in the membrane of cell bodies, as well as in the membrane of the dendrites (Fig 1B). Indeed, looking at Fig 1C it seems that there is more expression in the membrane of the dendrites than in the membrane of the cell bodies. This can be explained by the large surface area of the dendritic processes compared to the relatively small surface area of the cell body. Since the expression of membrane-bound ReaChR should be proportional to the membrane surface, more mCitrine fluorescence should be visible in the dendritic processes. It is also important to note that we did not enhance mCitrine with an antibody, thus the mCitrine signal is relatively weak in the membrane of cell bodies.

Concerning Fig 5A, we think that these different results are due to the different experimental conditions in mouse and macaque. In the mouse, ReaChR was expressed in vivo, while for the macaque explants we had to induce ReaChR expression ex vivo under cell culture conditions, within a few days. It is also possible that the cellular trafficking of the synthesized proteins could evolve over time and we might have not gotten the fully developed pattern of expression in our short window imposed by ex vivo experiments in the macaque retina.

Minor Corrections:

--Please explicitly state that MEA stands for multi-electrode array

We have updated our manuscript and incorporated this change.

--Typo: "To demonstrate the clinical translatability of our approach we tested if ReaChR can be functional, when expressed in RGCs of the primate retina."

--Fig1: it seems that B and C are mislabeled and should be switched.

We thank the referee for pointing out these mistakes. We have corrected the typo indicated and updated the figure legend in our manuscript.

Referee #2 (Comments on Novelty/Model System):

I am afraid that this work neither presents a novel molecular tool nor any progress to advance a potential optogenetic therapy for blindness.

Here my detailed arguments:

(a) ReaChR is an optogenetic tool that has been characterized in detail previously in vitro and in vivo (incl. spectral sensitivity and kinetics; Lin et al., Nat Neurosci, 2013)

ReaChR has never been tested in the retina, but characterizing a new optogenetic tool in a relevant tissue is of major importance, as expression levels are tissue dependent, which is a main parameter in determining sensitivity and kinetics.

(b) Optogenetic vision recovery is a blooming field and its feasibility has been shown multiple times in the blind rd1 mouse model (Tyagarajan et al, J Neurosci, 2010; Lagali et al, Nat Neurosci 2008; Cronin et al., EMBO Mol Med 2015, Macé et al. Science Transl Med, 2015, Gaub et al, Mol Ther 2015, etc.)

Yes, we agree. We have provided additional citations to refer to key studies in the field. Within our own group we have published previously on the targeting of optogenetic actuators to the photoreceptors (Busskamp, Duebel, Balya et al., Science 2010) as well as to bipolar cells (Macé et al., Mol Ther 2015). In the current study, we have targeted the ganglion cell population (as opposed to other cell types) with a red-shifted channelrhodopsin recognizing an unmet medical need for patients with advanced stage retinal disease.

(c) The state-of-the-art target cells for vision recovery in advanced stages of photoreceptor degeneration are not the RGCs that were chosen as targets in this study, but the retinal bipolar cells, the interneurons between photoreceptors and RGCs. Reason for this is that the intrinsic retinal signal computation can be mostly retained if targeting the Bipolar cells, in particular the important segregation of ON- and OFF pathways. This has been shown and argued for example in Macé et al., Mol Ther 2015, Gaub et al. Mol Ther 2015 and van Wyk et al PLoS Biol 2015. In contrary, when RGCs are targeted, it was shown that all RGCs turn into "ON-cells" (by the way, this important point has been completely neglected in this study), which was argued to be problematic (Tyagarajan et al, J Neurosci, 2010; Lin et al, PNAS 2008). As the genetic tools for bipolar cell targeting (promoters, AAVs) are available, the taken approach by Sengupte et al. appears as a step backward in development.

The reviewer's argument is based on a purely circuit based view of how visual restoration should be done. But what is important to note in a translational perspective is that the ideal cell targets for optogenetic therapy are determined by the available cells in patients. What cells can be targeted will be determined by the needs of patients which display various phenotypes going from patients with a large population of 'dormant' cone photoreceptors to patients with only RGCs remaining. Indeed, there are patients who display highly disorganized bipolar cell layer (Jacobson et al, 2012) for whom the only option would be to target RGCs.

The choice of targeting RGCs is further supported by clinical results from RGC stimulation with epiretinal implants. These patients can perform visual tasks, such as object localization, motion discrimination, and discrimination of oriented gratings (Humayun, Ophthalmology 2012; Shepherd, Trends Biotechnol 2013). Importantly, these studies indicate that the human cortex has the capability to adapt to a visual code that is different from the neural activity pattern conveyed by a normal retina.

In fact, today, the only FDA approved clinical trial in optogenetics aims to insert Chr2 into RGCs (www.blindness.org/RetroSense).

We agree that bipolar cells are an attractive target for optogenetic vision restoration, because neural circuits upstream of RGCs are utilized. However, as mentioned above, patients with compromised bipolar cell layers would not be eligible. Moreover, AAV technologies are at this stage inefficient in specifically targeting bipolar cells in non-human primates; although this has been accomplished in mice (Macé et al. Mol Ther 2015; Cronin et al., EMBO Mol Med, 2014; Doroudchi et al., Mol Ther 2011).

In conclusion, we think that there are different valuable therapy options, depending on the remaining cell-types in blind patients (i.e. 'dormant' cones or bipolar cells or RGCs). Thus, there is no such thing as "state-of-the art", when it comes to targeting various cell-types with optogenetic tools.

In our revised manuscript, we explain further the rationale for targeting RGCs in the discussion (2nd paragraph)

(d) Although the main message of this manuscript is that ReaChR can be activated by "safe" light intensities, ReaChR is by far not the first available optogenetic tool with which this is possible. In fact, the trend is to move away from channelrhodopsin variants towards vertebrate opsins, which are naturally 2-3 log units more light sensitive than channelrhodopsin, as they are G-protein coupled receptors and not simple ion channels. An additional advantage of mammalian opsins as potential therapeutic tools is the reduction of the possibility of an adverse immune response on long-term expression; this risk is a significant factor when contemplating the use of microbial channelrhodopsins in humans (and again, this important translational point has been completely

neglected in this study). Multiple opsin-based optogenetic tools have been shown to be at least equally if not more light sensitive than ReaChR, e.g. melanopsin in RGCs (Lin et al, PNAS 2008), Rhodopsin in BPCs (Gaub et al. Mol Ther 2015), Opto-mGluR6 in BPCs (van Wyk et al. PLoS Biol 2015). To exemplify this, melanopsin was shown to be activated at indoor lighting intensities and Opto-mGluR6 was shown to be fully activated at ambient daylight. Both would be much better suited for patient treatment than ReaChR, which would require, as stated by the authors in the discussion, the "patient to wear stimulation glasses in order to stimulate with appropriate light intensity".

Just like retinal cell-type targeting, the question of what optogenetic tool to use is currently an unsolved problem and we do not believe this choice should be dictated by "trends" but rather by scientific evidence weighing the pros and cons of each system for a particular application.

We agree that G-protein coupled vertebrate opsins are newly emerging and valuable alternatives to microbial opsins and we have now cited these recent articles in our revised manuscript.

Furthermore, we added a paragraph to the revised discussion where we examine the advantages and disadvantages of microbial opsins versus vertebrate opsins for optogenetic vision restoration.

(e) Gene therapy in the used model systems (rd1 mouse eye, macaque and human retina) has already been applied, even in the in vivo macaque eye in contrast to macaque retinal explants targeted in this study. Relating to the gene therapy approach, I came across some methodological contradictions of the authors themselves compared to their previous publications, for example:
- why did the authors in this study use lentiviruses and not AAVs to infect human retinas? AAVs are clinically favored vectors and the authors showed previously that they can successfully AAV-infect and maintain human retinal explants, which they are unable to do in this study for unknown reasons (Fradot et al., Hum Gene Ther. 2011)?

We are not aware of any published studies applying optogenetic therapy, in vivo, in the macaque eye. The goal of our experiments was to demonstrate that ReaChR is functional in human RGCs, when it is expressed ex vivo under the same promoter (hSyn) that we used in vivo in mice. We chose to use lentivirus, which has the advantage of leading to very fast (<2 days) and robust expression in explants. It is important to note that the human post mortem retinas stay physiologically viable for a much shorter time than the monkey retinal explants. This is due to the significant post mortem delays involved, when they were isolated from the donors. We thus produced the lentivirus with the sole objective of using it for the human retina considering the importance of getting ReaChR expressed as fast as possible in order to be able to perform physiological recordings. Lentivirus was used as well in previous studies using optogenetics to activate human retina (Busskamp et al., 2010). In the work by Fradot et al, it is clearly stated that expression took >2 weeks and native YFP expression was undetectable until following immunohistochemistry at one month. We confirm that lentivirus is not useful for future in vivo studies, due to physical access barriers to the retina with this larger sized particle. This is why we also tested if ReaChR is functional in macaque explants, when AAV is used. In order to accelerate expression in the ex vivo macaque retina, we switched to a stronger and faster promoter (CAG) instead of changing the vector.

- why did the authors use an AAV2/8(Y447,733F) capsid variant when they had recently shown that the Y-F capsid mutant AAVs are less efficient in retinal transduction compared to their own AAV2/2(7m8) capsid variant that was shown to transduce the macaque retina very well (Dalkara et al., Science Transl Med 2013)?

This is again a methodological choice. We confirm that AAV2-7m8 outperforms AAV2-4YF mutant in vivo looking at general intensity and depth of retinal transduction. However, both AAV2 based variants suffer from the slow kinetics of AAV2 mediated gene expression compared to the faster unpacking AAV8- which was chosen in our ex vivo experiments to respect time constraints imposed by organotypic cultures. AAV8 was further strengthened by the addition of two tyrosine mutations.

(f) The behavioral paradigm used (open field test) is weak and does not comply with state-of-the-art. The open-field test only shows that the mice react to light (a behavior known to be mediated by intrinsically photosensitive RGCs alone), but it does not define the limits of the visual abilities provided by the optogenetic therapeutic. A water maze, for example that was used in many of the

above-mentioned studies, would have been a suited experiment to see whether these mice can carry out visual discriminations with a cognitive component. In particular, behavior should have been tested for light sensitivity as it was done for example in van Wyk et al., PLoS Biol 2015.

We have performed an additional behavioral task, the light-dark box test, and demonstrate that ReaChR treated rd1 mice exhibit light aversion and are capable of distinguishing light from dark (Fig 4C and D). Moreover, we have tested the light sensitivity of the mice in this test (Fig. 4C). Aversion to light is an innate behavior in sighted mice mediated by the limbic neural circuitry (reviewed in Calhoun, Tye, Nat Neurosci 2015). This behavior is abolished in rd1 mice. Note that the light-dark chamber preference of the rd1 mice in our study did not change for a wide range of stimulus intensities, in stark contrast to the wild type mice. Restoration of this behavior suggests that the limbic neural circuits might be able to utilize the light driven signals originating from the retinas of the ReaChR- treated mice.

We acknowledge in the discussion that this is a test of light sensitivity but does not measure sophisticated visual function.

We think that visual function for the RGC targeted optogenetics approach would be best studied in AAV injected macaques in a future study and our manuscript fits as a gateway to such future studies.

(g) Milestone references in the field are not indicated - this puts a misleading light on the novelty of this work, particularly as the discussion is massively overstated.

These papers are now cited: Wyk et al., PLoS Biol 2015; Gaub et al., Mol Ther 2015, Cehajic-Kapetanovic et al. Curr Biol 2015; Doroudchi et al., Mol Ther 2011; Cronin et al. EMBO Mol Med 2014; Macé et al. Mol Ther 2015. In the discussion, we refer to these publications, and we acknowledge that there are important alternatives concerning the choice of cell-type (RGCs vs. bipolar cells) as well as the choice of optogenetic tool (microbial opsins vs. vertebrate opsins). Lastly, we have also toned down the discussion.

(h) The C3H/HeN rd1 mouse line is, as commonly known, not an optimal mouse model for vision recovery studies, as it carries the nob5 (GRP179) mutation that renders ON-Bipolar cells non-functional. C3H/HeOu for example, an alternative rd1 mouse model, does not carry the nob mutation.

In our study we target ganglion cells and any additional mutations upstream of these cells will not have an effect on the results. This GPR179 mutation, upstream of our activated cells, is thus completely irrelevant and rd1 the most frequently used model in vision restoration studies.

(i) the Methods part "calculation of safety threshold for optogenetic stimulation" is copied 1:1 from "Degenaar et al., J Neural Eng, 2009" but does not apply to the experimental settings of this study! The authors work on retinal explants and not on human patients, but the formulas all include light attenuation factors through the optic apparatus. Therefore, the "safe" light intensity on a directly illuminated retinal explants would be at least 3 log units lower than claimed by the authors.

From Degenaar et al (2009), we used the formula that relates retinal irradiance to the radiance of a light source. Indeed, this formula includes light attenuation factors through the optic apparatus. Because we used retinal explants, we needed this formula to relate the light intensities we used (expressed in terms of direct retinal irradiance) with the light intensities cited by the International Commission (expressed in terms of source radiance). In fact, this formula allowed us to apply the safety threshold stated by the International Commission to this study.

Please note that the calculation of retinal irradiance has been described before by Sliney et al. in 1995. Please see: Sliney, D.H., P. Fast, and A. Ricksand, Optical radiation hazards analysis of ultraviolet headlamps. Appl Opt, 1995. 34(22) Page: 4914. Therefore, we have cited both Sliney et al. 1995 and Degenaar et al. 2009. Please see also: International Commission on Non-ionizing Radiation Protection (ICNIRP) On limits of exposure to incoherent visible and infrared radiation. Health Physics, 2013. Page: 79 (see also page: 84 table2; page: 87 eqns 10, 14). There, the limits of irradiance on the retina are clearly described.

(j) Many of the chosen methods are "outdated" and don't give novel insights to move the field forward (see also Reply to Authors for details).

(k) The optogenetic vision recovery field is progressing rapidly and translationally oriented studies have to compare different optogenetic approaches under the same conditions, i.e. investigate the validity of viral vectors (e.g. capsid engineered AAVs) & specific promoters from the mouse in macaque and human tissue, compare and quantify expression levels and biodistribution of the optogenetic therapeutic and determine the potential quality of recovered vision. As none of these issues have been addressed in the present study, I do not recommend this submission for publication in the highly ranked journal EMBO Mol Med.

Our study is not intended towards comparing all existing optogenetic approaches for visual restoration but rather aims at generating a full data set on the suitability of ReaChR for this application, namely from the angle of light safety.

Referee #2 (Remarks):

This manuscript by Abhishek Sengupta et al. describes the use of the red-shifted Channelrhodopsin, ReaChR, to recover light-sensitivity in the retinas of blind mice as well as in macaque and human retinal explants. For this, ReaChR has been introduced into the retina with AAV and lentiviruses under ubiquitous promoters, nonetheless, electrophysiological characterization has been restricted to the retinal ganglion cells (RGCs).

While I appreciate the use of 3 model systems, the mouse, macaque and human, I am somehow disappointed that the potential of these systems to test for translatability was not exploited, e.g. the authors did not test for the specificity of available promoters, the efficiency and biodistribution of AAV transduction, the time to transgene expression, determination of the quality of the potentially recovered vision, etc. In actual fact, the choice of experiments and the manuscript text appear somehow elusive and the data analysis and discussion somehow superficial. I am afraid that the line of argumentation is most surely not understood by a non-expert, as the reasoning is incomplete.

The investigation of all the parameters listed above (specificity of available promoters, the efficiency and biodistribution of AAV transduction, the time to transgene expression, determination of the quality of the potentially recovered vision) in mouse, macaque and human, go beyond the scope of our current manuscript. Our study aims at generating a full data set on the suitability of ReaChR for this application, namely from the angle of light safety. We have taken into account the criticism of the reviewer and we made efforts to clarify this point in our revised manuscript.

Major Points:

1. The main argument of this study is the "safe" light-activation of ReaChR. ReaChR is, however, by far not the first available optogenetic tool to function in the "safe light intensity" range. In fact, the trend is to move away from ChR2 variants towards vertebrate opsins, which are naturally 2-3 log units more light sensitive, as they are G-protein coupled receptors and not simple ion channels. An additional advantage of mammalian opsins as potential therapeutics is the reduction of the possibility of an adverse immune response on long-term expression; this risk is a significant factor when contemplating the use of microbial channelrhodopsins in humans. Multiple opsin-based optogenetic tools have been shown to be at least equally if not more light sensitive than ReaChR, e.g. melanopsin in RGCs (Lin et al, PNAS 2008), Rhodopsin in BPCs (Gaub et al. Mol Ther 2015), Opto-mGluR6 in BPCs (van Wyk et al. PLoS Biol 2015). To exemplify this, melanopsin was shown to be activated at indoor lighting intensities and Opto-mGluR6 was shown to be fully activated at ambient daylight. Both would be suited for patient treatment. In contrary, ReaChR, as stated by the authors in the discussion, would require the "patient to wear stimulation glasses in order to stimulate with appropriate light intensity". Above references have to be added to the manuscript and discussed in comparison to ReaChR. Also, the argument of the novelty of "safe light activation" has to be tuned down throughout the manuscript.

Again, we agree that vertebrate opsins are newly emerging and valuable alternatives to microbial opsins, but we do not believe the choice of an optogenetic tool should be dictated by "trends" but rather by scientific evidence weighing the advantages and disadvantages of each system for a particular application. Therefore, we added a paragraph where we discuss the advantages and

disadvantages of microbial opsins versus vertebrate opsins for optogenetic vision restoration in our revised manuscript. Please see also our answer to the reviewers comment (d).

2. On the same note: I actually doubt that ReaChR allows for potential treatment with "safe" light intensities, and this is why: The EU Directive was developed for human individuals being exposed to radiation. Formula (2) on p7 includes light attenuation through the optic apparatus, which Degenaar et al., J Neural Eng, 2009 included in their analysis. The authors here, in contrary, work with retinal explants. Therefore, the "safe light intensity threshold" for retinal explants would be shifted to approximately 3 log units lower light intensities. In other words, ReaChR is not "safe". As there seems to be some confusion about light intensities, I would like the authors to include actual values into the formulas for their experimental settings and relate these to the results.

Indeed, this formula includes light attenuation factors through the optic apparatus. Because we used retinal explants, we needed this formula to relate the light intensities we used (expressed in terms of direct retinal irradiance) with the light intensities cited by the International Commissions (expressed in terms of source radiance). In fact, this formula allowed us to apply the safety threshold stated by the International Commission to this study. Please see also our response to the reviewer's argument (i).

3. As argued by several recent papers (MacLeod et al, Mol Ther 2015, Gaub et al. Mol Ther 2015 and van Wyk et al PLoS Biol 2015), the state-of-the-art target cells for optogenetic vision recovery in advanced stages of photoreceptor degeneration are not the RGCs, but the ON-bipolar cells. Reason for this is that the intrinsic retinal signal computation can be greatly retained when the ON-bipolar cells are targeted, in particular the important segregation of ON- and OFF pathways. In contrary, when RGCs are targeted, it was shown that they all turn into "ON-cells" and that the "transientness" of their response variety is reduced (Tyagarajan et al, J Neurosci, 2010; Lin et al, PNAS 2008). Both of these points, despite their importance for the quality of recovered vision, have been entirely neglected in this study - what percentage of RGCs expressed ReaChR? Did other cells also express (due to the ubiquitous promoters used)? What was the RGC response variety in treated rd1 compared to healthy retinas (ON-, OFF-, transientness)? What are the potential consequences on recovered vision if for example only sustained ON-responses are being recovered? As the genetic tools for bipolar cell targeting (promoters, AAVs) are available, I do not understand why the authors targeted the RGCs in this study. All of the above has to be shown, analyzed and discussed in the manuscript.

Retinal cell-type targeting by optogenetics is at present a largely unsolved problem. Thus there is no such thing as state-of-the art, when it comes to targeting various cell types with optogenetic tools in retinitis pigmentosa. What cells can be targeted will be determined by the needs of patients which display various phenotypes going from patients with a large population of dormant cones to patients with only RGCs remaining. Please see also our response to the reviewer's argument (c).

4. The C3H/HeN rd1 mouse model is not well suited for vision recovery studies as it is known to possess the nob5 (GRP179, no b-wave) mutation, which renders the ON-Bipolar cell pathway non-functional (Nishiguchi KM et al., Nat Communications, 2015). Vision rescue is therefore only achievable through the OFF-pathway. Is this why the RGCs and not the Bipolar cells have been targeted by the authors? If yes, this has to be mentioned in the text.

In our study we target ganglion cells and any additional mutations upstream of these cells will not have an effect on the results. This mutation upstream of our activated cells is thus completely irrelevant and rd1 is the most frequently used model in vision restoration.

5. Gene therapy: I came across some methodological contradictions of the authors themselves compared to their previous publications, for example:
- why did the authors use lentiviruses and not AAVs to infect human retinas? AAVs are clinically favored vectors and the authors showed previously that they can successfully AAV-infect and maintain human retinal explants for sufficient time to expression (Fradot et al., Hum Gene Ther. 2011)?

The goal of this experiment was to demonstrate that ReaChR is functional in human RGCs, when it is expressed ex vivo under the same promoter (hSyn) that we used in vivo in mice. We chose to use

lentivirus, which has the advantage of leading to very fast (<2 days) and robust expression in explants. Lentivirus was used as well in previous studies using optogenetics to activate human retina (Busskamp et al., 2010). In the work by Fradot et al., it is clearly stated that expression took >2 weeks and native YFP expression was undetectable until following immunohistochemistry at one month. We confirm that lentivirus is not useful for future in vivo studies, due to physical access barriers to the retina with this larger sized particle. This is why we also tested if ReaChR is functional in macaque explants, when an AAV vector. In order to accelerate expression in the ex vivo macaque retina, we switched to a stronger and faster promoter (CAG) instead of changing the vector.

- why did the authors use an AAV2/8(Y447,733F) capsid variant when they had recently shown that Y-F capsid mutant AAVs are less efficient in retinal transduction compared to their own AAV2/2(7m8) capsid variant that was able to transduce the macaque retina very well (Dalkara et al., Science Transl Med 2013)? These points have to be discussed; as the AAV2/2(7m8) variant emerged from the author's labs, I would like to see at least an AAV2/8(Y447,733F) and an AAV2/2(7m8) transduced macaque (and human) retina side by side. I would also like to have a clear explanation on how long the human retinas could be maximally cultured and why the authors did not achieve the results they described earlier in Fradot et al (2011).

Please see below images comparing in time AAV2-7m8 versus AAV8-2YF in primate explants. Please notice the difference in kinetics justifying the use of AAV8-2YF under these specific conditions. We do not argue that this would be our vector of choice for in vivo experiments in primates.

AAV2-7m8-GFP

Time — Day 4 — Day 6 —>

AAV8-2YF-GFP

Fradot et al. did not demonstrate any optogenetic activity mediated by ChR2 in organotypic cultures. In fact, no electrophysiological recordings were presented in this manuscript. This was likely due to the extended time periods needed to get sufficient ChR2 expression and the poor state of the retina at the time sufficient expression was obtained. In our hands, we have tested and obtained spontaneous spike responses in the human retina up to 5 days; longer incubation periods were not tested.

Here, we have provided a functional response profile of our candidate optogenetic tool in the macaque and human retina by directly recording from them electrophysiologically. This is a promising indication of its functionality in primate tissue.

6. Experiments on Primate Retina: I have a few comments and the methodology leaves many open questions:

- Ref 22 (Fradot et al) explains the methodology for the human retinal explants, but a much better Ref, including AAV transduction, parafoveal expression and activity patterns in midget RGCs would be "Yin L et al., J Neurosci 34, 2014"

This reference is also cited.

- it is not explained why the CAG promoter instead of the hSyn promoter was used; maybe because Yin et al. (2014), Dalkara et al (2013) and others used the CMV and CAG promoters to transduce RGCs? This should be reasoned and either identical promoters should be used in all three models or the different promoters should be compared in their efficiency and distribution of expression.

In general, cell-type specific promoters are weaker and slower than ubiquitous promoters; this is why we chose to use CAG to get the maximum expression in the shortest time frame in retinal explants. Please see also our answer to the reviewers comment (e).

- I do not believe that the authors saw expression already 3d after AAV transduction; typically AAV2/2 takes 3-4 weeks to express, AAV2/8 might be slightly faster (Natkunarajah M et al., Gene Ther, 2008. Please adapt the text accordingly.

The question of how long it takes to get expression with a given vector depends on administration route, dose and promoter. Our ex vivo conditions here cannot be compared to in vivo studies such as the one the referee cites. Under conditions we optimized –including very high particle number that might not be compatible with in vivo experiments- and we are able to get expression in a matter of 3 days in primate explants (please see above data).

7. In my opinion, the chosen behavioral paradigm, the open field test, does not suffice as a stand-alone behavioral test. Whilst the open-field test is an established test for photophobia and shows if the mice react to light (a behavior that is known to be mediated by ipRGCs alone, see e.g. Semo M et al., PLoS ONE 2010), the test does not show any visually-guided behavior and does not define the limits of the visual abilities provided by the optogenetic therapeutic. A water maze, for example, as used in many previous studies, would have been a suited experiment to see whether these mice can carry out visual discriminations with a cognitive component, for example distinguish between distinct light patterns (moving versus static patterns, etc.), or simply what acuity the recovered vision has (OKR). Besides additional behavioral paradigms, I would also like to see the "light sensitivity of the behavioral response", as it was shown in van Wyk et al., PLoS Biol 2015, as this is the "real" readout for sensitivity.

We performed additional experiments in order to measure the "light sensitivity of the behavioral response". Therefore, we used the light-dark box with different light intensities. Please see the updated figure (Fig 4 C, D) for the behavioral tests.

Concerning the potential effect of ipRGCs: The untreated rd1 control mice have functional ipRGCs but they did not show any response to light (590 nm), even at the highest light intensity. This can be explained by the fact that the sensitivity of melanopsin at this wavelength is extremely low, compared to its optimal excitation wavelength (blue light at 479 nm). The observation that orange light had no effect on ipRGCs is also consistent with our results from the cortical recordings where rd1 control mice did not show any response to orange light at 590 nm.

8. I am missing a figure and text describing the variety of RGC output, e.g. a statement about the transientness of RGC responses, if all RGCs were turned into ON cells by ReaChR expression (as expected) and what consequences this may have for the potentially recovered vision. Also, I would like to know how many RGCs expressed ReaChR, how the exact biodistribution of expression looked like was and how this influenced the locally measured RGC signals. I would also like to see these points discussed and compared between the 3 model systems (mouse, macaque, human).

When RGCs are stimulated directly, then it is obvious, that all "RGCs are turned into ON cells". We also confirm that the segregation of transient and sustained RGC responses cannot be retained when RGCs are stimulated directly. Naturally, this is always the case, when RGC stimulation approaches are used, regardless of the stimulation technique (i.e. optogenetic stimulation or electrical stimulation) (Thyagarajan et al., J Neurosci 2010). Our choice of targeting RGCs is

supported by clinical results from RGC stimulation with epiretinal implants. Even though these implants activate both ON- and OFF RGCs at the same time, patients – implanted with the Argus II epiretinal system – were enabled to perform visual tasks with an acuity up to 20/1262 (Snellen equivalent) (Humayun, Ophthalmology 2012)(Shepherd, Trends Biotechnol 2013).

Please see also our answer to the reviewers comment (c).

The comparison of the biodistribution of expression in all 3 model systems (mouse, macaque, human), and how this influenced the locally measured RGC signals, go beyond the scope of our current manuscript.

9. Experiments on human retina: Why is expression only seen in mid-peripheral RGCs and not also in peripheral RGCs? This makes no sense as the lentivirus was applied panretinally. Additionally, the fovea/parafoveal region may present a "problem zone" for optogenetic vision recovery, as the RGCs are skewed to the side, away from the fovea towards the parafovea; you should hypothesize and discuss what this may mean in terms of recovered vision and a potential therapy-skewing of the picture that requires correction optics?

We did not claim that expression was restricted to mid-peripheral RGCs. In fact, expression was seen both in central RGCs and in peripheral RGCs. This is what we show in Fig. 7 A (RGCs located ~500 μm from the center) and Fig. 7 B (RGCs located 3-4 mm from the center). Now, we state this more clearly in the legend of figure 7, and in the results.

In fact, the parafoveal region is a desired zone and not a "problem zone" for optogenetic vision recovery; and this is why: RGCs in the central retina have a much smaller dendritic field size than RGCs in the periphery. Hence, optogenetic stimulation of the very small dendritic fields (ca 5-10 μm in diameter) allows for stimulation at much higher spatial resolution, compared to stimulation of the relative large dendritic fields of peripheral RGCs (>100 μm in diameter).

Finally, there is a high chance that treated patients could make good use of the visual information that is transmitted from ReaChR-expressing RGCs to the visual cortex, even though these RGCs are shifted away from the fovea. In particular, clinical studies with retinal implants indicate that the human cortex has the capability to adapt to a visual code that is different from the neural activity pattern conveyed by a normal retina.

It is important to note that not only RGCs but also the ON-bipolar cells are "skewed to the side, away from the fovea". Hence, there is the same problem of "distortion" when bipolar cells of the central human retina are targeted with an optogenetic tool.

Minor Points:

1. Although I appreciate that the authors went through the trouble measuring the action spectrum of ReaChR in each setting, I do not understand what their motivation was to do so. The ReaChR action spectrum has been determined in the original publication by Lin et al. (Nat Neurosci 2013) and would not change if nothing is engineered on the protein. Fig 4E for example would totally suffice.

We think that it was important to measure the action spectrum directly in the retina. In fact, we observed that the peak activity of ReaChR (ca. 550 nm) was slightly shifted towards shorter wavelengths compared to the peak sensitivity of 590 nm described by Lin et al. (Nat Neurosci 2013). Our observation is in line with other studies from Dawydwow et al., Proc Nat Acad Sci 2014 (doi/10.1073/pnas.1408269111) and from the Boyden lab (Klapoetke et al. 2014 doi:10.1038/nmeth.2836) where the action spectrum of ReaChR was found to be shifted towards shorter wavelengths as well.

2. As ubiquitous promoters have been used, it is expected that ReaChR is expressed in other cell types than RGCs. Have the authors observed an "expression leak"? This point has been entirely hushed up in this study and must be investigated, stated and discussed.

hSyn in combination with AAV2 will lead to relatively specific expression in RGCs as previously published (Caporale et al., Mol Ther, 2012, Please see Figure 1 and Supplemental figure 1). In

agreement with previously published studies we did not observe any leak into other inner retinal cells.

3. p4, 3rd paragraph: it should be mentioned why input from ipRGCs is not of concern (Semo M et al., PLoS ONE 2010).

We used orange light at 590 nm. The sensitivity of melanopsin at this wavelength is extremely low compared to its optimal excitation wavelength (blue light at 479 nm). Thus, we have no effect of orange light (590 nm) at the light intensities that we used in our behavioral tests. This is in line with our observation that the untreated rd1 control mice did not show any behavioral response to orange light, and they do have ipRGCs. In addition, rd1 control mice did not show any response to orange light at 590 nm at the cortical level.

4. p.4, 4th paragraph: it must be explained why the promoter was changed from hSyn to CAG - is there any indication that CAG is better or hSyn does not function in the primate?

5. p4, 4th paragraph: is this a spelling error or was expression really observed after 3 days? While I would believe this is true for lentiviral expression, AAVs take in the best case a minimum of 2 weeks to express to my knowledge.

The question of how long it takes to get expression with a given vector depends on administration route, dose and promoter. Our ex vivo conditions here cannot be compared to in vivo studies such as the one the referee cites. Under conditions we optimized –including very high particle number that might not be compatible with in vivo experiments- and we are able to get expression in a matter of 3 days (please see above data) in primate explants. Please see our response to the referee's remark "6. Experiments on primate retina"

6. How do you explain the "uneven" and restricted ReaChR expression pattern in the retinas - centered expression in the mouse retina (e.g. Fig. 1A) and the parafovea-restricted expression in the human explants (e.g. expanded view figure 4)?

We did not claim that expression was restricted to the parafovea. ReaChR expression was seen both in central RGCs and in peripheral RGCs. This is what we show in Fig. 7 A (RGCs located ~500 μm from the center) and Fig. 7 B (RGCs located 3-4 mm from the center). Now, we state this more clearly in the legend of figure 7, and in the results. Please see our response to the referee's remark "9. Experiments on human retina"

7. The labeling of the "expanded view" figures appears to be wrong? For example, expanded view figure 4 corresponds to figure 6.

The "expanded view" figures have their own numbering order. We have cited them within the manuscript text according to their correct number.

We have renumbered the movies as Expanded View Movie Fig1–3.

8. Human retina: (1) why was only 1 retina further investigated when 8 retinas (from 4 donors) were available? I would have liked to see if the observations from the 1 retina can be generalized. (2) How do the authors explain expression to be restricted to mid-ganglion cell RGCs of the parafovea? (3) Why did the authors use lentiviruses for transduction, as they had previously shown that they can use AAVs (Fradot et al.) which is in light of translatability a much better suited vector? (4) The text on p.5, para 2 is not well written, the procedure and argumentation for all the taken restrictions is unclear. (5) Contradiction: "identify retinas exhibiting persisting RGC spiking activity" ↔ "Note that the human retina was devoid of innate light responses at the time of isolation" (6) As the RGCs are skewed in the parafoveal region (as there is no RGCs directly below the fovea), what do you think will be the consequences on the quality of recovered vision for a ReaChR-treated patient? Would this skewing have to be corrected for optically?

(1) We would like to provide a couple of clarifications on the issue of the availability of the human retinas. Although we received eyes from four deceased donors, we only received one eye from each subject while the other eye went to other groups for unrelated experiments. We want to stress here

that a physiologically viable retina from deceased human subjects is incredibly difficult to obtain, in large part due to the significant delays involved in receiving the eye post mortem. As we state in the manuscript, we used the MEA to identify retinas where retinal ganglion cells were still physiologically viable (based on the presence of spontaneous spiking activity). Out of the four retinas we tested, we found spontaneously spiking ganglion cells in only one retina out of the four we tested. This particular retina was also the one we received with the least post mortem delay (10 h). We prepared explants from this retina, transduced it with lentivirus and performed physiological recordings. Groups elsewhere may have quicker access to post-mortem human retinas but a survey of the literature indicates that physiological data recorded directly from the human retina is rare.

(2) Please see our response to the referee's remark "9. Experiments on human retina"

(3) Please see our response to referee's comment (e)

(4)/(5) There is no contradiction here. Spontaneous spiking by retinal ganglion cells is a fundamental feature of the retina across all species. While the frequency and patterns of spontaneous firing differ from cell to cell, these spontaneous action potentials are not elicited by light. Thus when we state that the human retina was devoid of light responses at the time of isolation we imply that light induced spiking was absent. The spontaneous spiking of the retinal ganglion cells in the human retina (at the time of isolation) could not be correlated with the onset or offset of light stimulation. We exploited the property of spontaneous spiking in the retina to identify retinas with physiologically viable retinal ganglion cells. A retina with physiologically active ganglion cells, firing spontaneous action potentials, was chosen for expressing ReaChR.

(6) Please see our response to referee's remark "9. Experiments on human retina"

9. The discussion overstates at multiple places, please lessen your arguments. For example: (1) "excellent vision restoration tool" - ReaChR was only shown to recover light responses, but not "vision", (2) "temporally sufficient retinal responses" - for what sufficient? The open-field test does not require great temporal or spatial resolution. Such a strong argument should be shown behaviorally. (3) "compelling responses at the behavioral level" - actually, only a very basic behavioral paradigm has been used that does not show "behavioral vision" (4) "expression without cytosolic aggregation" - from what you show I see many intracellular aggregations and surprisingly weak labeling of the cytoplasm around the soma. Do you have photomicrographs that agree better with your statement? (5) "we could demonstrate clinical translatability" - I disagree, to foster translatability you should have analyzed expression efficiency, biodistribution, cell specificity, promoter, viral validity in the models in parallel, quality of recovered vision, etc. (6) "the treatment we propose has a high potential to be rapidly translated as a therapeutic approach for clinical trials" - I again disagree, there exist more promising optogenetic approaches and nothing has been shown in this study that moves translatability forward. (7) "first proof of principle that with an optogenetic approach it should be possible to stimulate midgenet RGCs" - please tone this down, as it was shown previously that AAVs transduce midgenet RGCs in primates with an optogenetic tool (GCaMP5 ; Yin et al, 2011 and 2014). A bigger concern to me is the fact that you seem to only have transduced this RGC cell type, although you applied the lentivirus panretinally.

We rewrote the discussion, toned down the criticized over-reaching statements, and we acknowledge that there are valuable alternatives to our vision restoration approach, such as the use of vertebrate opsins and the targeting of bipolar cells.

Referee #3 (Comments on Novelty/Model System):

This is state-of-the-art research in use of channel opsins to restore vision-like behavior

Referee #3 (Remarks):

This is a most interesting manuscript describing the use of a red-shifted channel opsin for restoring light responses to blind retinas in mice and to pharmacologically blinded retinas in primates including humans. This work is important as the use of a red-shifted channel opsin allows higher light intensities to be used safely for activation; this overcomes the challenge of the original (blue) channel opsins which could only be activated at potentially toxic levels of light.

Major comments:

These experiments are overall well designed and executed.

The behavioral assay is relatively crude and measures sensing of light rather than vision per se. The authors should acknowledge this limitation. Did they perform any two choice forced tests or other more visual tests on these animals?

We thank the reviewer for his encouraging comments and appreciate the constructive criticism on the choice of behavioral experiment. For our revised manuscript, we performed an additional behavior test, the light-dark box test (Fig. 4 C, D). In this experiment we determined the light intensity that is required for ReaChR treated rd1 mice to distinguish light from dark (Fig 4C and D).

In the discussion of the revised manuscript, we acknowledge that this is a test of light sensitivity but does not measure sophisticated visual function. We have also updated throughout the text (including the title) that the ReaChR responses are light responses.

The paper would be improved by a figure panel for both macaque and human retinas showing MEA recording of non-transfected retinas under L-AP4 blockade to demonstrate complete absence of firing.

We thank the reviewer for this helpful suggestion. We now provide a representative voltage trace of an uninfected macaque retina under L-AP4 (Expanded View Fig 3C) in our revised manuscript. We tried this protocol in three different macaque retinas (2 animals) and found that bath application of L-AP4 completely blocked endogenous light responses. We could not perform this on human retinas due to non-availability of viable retinas from post mortem donors.

The difference between MEA and patch clamp 'action spectra' is interesting. To what do the author attribute this? Could this be due to ipRGCs detected on MEA skewing the spectrum to the blue?

We think that the slight difference between patch-clamp and MEA results can be explained by the different recording techniques: In our patch-clamp experiments, we measured ReaChR-photocurrents, whereas in our MEA recordings we recorded the spiking output. The photocurrents induced by ReaChR lead to depolarization of the membrane potential. This depolarization then activates voltage-gated sodium-channels which in turn elicit spike responses, after reaching the spike threshold. In our patch-clamp experiments, cells were voltage clamped at -60mV , and lidocaine (blocking Na-VGCs) was present in the intracellular solution. Thus, we recorded 'pure' photocurrents with patch-clamp; while with MEA recordings we observed directly the RGC-spiking activity. From the 'action spectra' figure we may argue that the transformation of the ReaChR-photocurrents into spiking activity is not linear.

Although we cannot rule out an effect of ipRGCs, we think that they play minor role. In MEA experiments we observed a vast majority of transient responses (shorter response than stimulus duration). Moreover, all the cells displayed short latencies, which is incompatible with the slow and sustained responses of ipRGCs.

To that point, what the authors present as action spectra appear to be single-fluence relative activity spectra. A true action spectrum requires 1.) performing a complete irradiance response relationship for each wavelength tested, 2.) demonstrating that these are all fit by the same curve (Hill coefficient or Michaelis-Menten curve) and then 3.) plotting the 150 point for each wavelength. Only if all curves have the same form does the principle of univariance hold. The authors should clarify that their curves are 'single fluence relative action spectra' if this is indeed the case.

We thank the reviewer for bringing this point to our attention. We used different wavelengths at a constant light intensity for these measurements. So it is correct, that we present "single-fluence-relative action spectra". This intensity was carefully chosen at $\sim 10^{16}$ photons $\text{cm}^{-2} \text{s}^{-1}$ where the photon flux is high enough to elicit robust responses, but well below the safety threshold. This point is now clarified in the revised version of the manuscript.

Finally, several recent studies have reported remarkable vision restoration using AAV-delivered rhodopsin to ON-bipolar cells (Gaub et al., Mol Ther. 2015 Jul 3. doi: 10.1038/mt.2015.121; Cehajic-Kapetanovic et al., Curr Biol. 2015 Aug 17;25(16):2111-22. doi:

10.1016/j.cub.2015.07.029). It appears the more 'upstream' expression of the opsin restores more vision-like firing behavior to RGCs. The authors may wish to discuss whether their red shifted channel opsin could be targeted in this way to bipolar cells.

Yes, we agree that bipolar cells are an attractive target for those blind patients where the inner nuclear layer and the synapses between bipolar and ganglion cells remain intact. However, it is important to note that vector promoter combinations for specifically targeting bipolar cells in higher primates do not exist at this point in time (although we are able to target this cell population nicely in mice- please see our recent paper Macé et al., 2015, Mol Ther as well as work by Cronin et al. EMBO Mol Med 2014, Doroudchi et al Mol Ther 2011).

We now evoke these points in our revised discussion section (page 6): "In general, one should keep in mind that the choice of cell-type depends on the availability of surviving retinal cells. Hence, those patients where the inner nuclear layer and the synapses between bipolar and ganglion cells remain intact, could be eligible candidates for a bipolar cell based optogenetic treatment with ReaChR (Jacobson et al. 2012). The feasibility of targeting channelrhodopsin variants specifically to ON-bipolar cells via AAV-vectors has been demonstrated in mouse models of retinal degeneration (Macé et al., Mol Ther 2015, Cronin et al., EMBO Mol Med 2014, Doroudchi et al., Mol Ther 2011), although this has not yet been accomplished in non-human primates. Thus, targeting of ReaChR to retinal ON-bipolar cells, that utilize neural circuits upstream of ganglion cells, could be a focus of future studies."

Minor suggestions:

- Were animals injected bilaterally or unilaterally?

Animals were injected bilaterally, we have clarified this in the methods.

- In figure EV1 it is mentioned that cells are visualized through the linked mCitrine fluorescent protein, but in panel B it is noted that a GFP-positive cell is being targeted. To be clear, is this an mCitrine-positive cell? Or is GFP used in some cases?

We thank the reviewer for bringing this error to our attention. We have updated the figure legend to state that an mCitrine-positive cell was targeted.

- A reference should be provided for the retinal tropism of AAV8 Y447 733F

We now added the following reference: Kay et al. Targeting photoreceptors via intravitreal delivery using novel, capsid-mutated AAV vectors. PLoS One. 2013 Apr 26;8(4)

- In the legend of Figure 1 it appears panels B and C are switched

We have corrected this in the figure legend.

2nd Editorial Decision

12 February 2016

Thank you again for the submission of your revised manuscript.

I have now received the evaluations from the two of the three Reviewers who were asked to re-review it. In fact, Reviewer 3, who was globally positive, was not reachable.

As you will see, while Reviewer 1 is now satisfied, Reviewer 2 still has concerns that s/he feels have not been satisfactorily addressed.

Reviewer 2 is still not convinced of the fundamental novelty of your work or of the potential advantage of the superiority of optogenetic RGC-targeted vision recovery versus retinal implants. S/he also disagrees on some of the chosen experimental approaches and is not satisfied with the additional behavioural test. The reviewer also lists other elements of concern.

Although as you know, it is EMBO Molecular Medicine policy to allow a single round of experimental revision only, after further discussion, I have decided to allow you to submit a re-revised version with the following requests. Specifically, I would invite you to respond point-by-point to the Reviewer's concerns and introduce the required textual changes for clarification and to reduce over-statements. Experimentally, I feel that to provide an additional cognitive-based vision test would be unnecessary, given that the light/dark box test was satisfactory for Reviewer 1 and that it would also imply significant time and use of animals, with ultimately unclear relevance for

primates. I would however encourage you to perform additional experiments on the non-human primate retinas to address the vector/promoter issues, ideally also on human material if available.

Finally, I also suggest that you carefully adhere to our guidelines for publication (<http://embomolmed.embopress.org/authorguide>) in your next version, including our new requirements for supplemental data (see also below) to speed up the pre-acceptance process in case of a positive outcome.

REFEREE REPORTS

Referee #1 (Comments on Novelty/Model System):

This manuscript has been very well revised. It was a pleasure to read. It is a very important contribution to our knowledge of optogenetic therapy of blindness, and the findings have major clinical implications.

Referee #2 (Comments on Novelty/Model System):

The only novelty of this work is MEA recordings from one human retinal explant expressing an optogenetic activator. While it is nice to see that cultured human retinas seem to be viable for some days, no new insight on the translatability of the molecular tools (such as viruses and Promoters) is given in this work nor any new insights on the goodness of optogenetic retinal Ganglion cell targeted vision recovery compared to retinal implants.

Referee #2 (Remarks):

Thank you for the extensive explanations and revisions. Unfortunately some points still remain that I would like to see addressed. I refer to the points of the initial review:

(1),(d),(4): Why to use ReaChR and what cells to target (paragraph 2 Discussion)

- The authors argue that RGCs are the only target for patients with highly disorganized or missing bipolar cell layer. However, neither the cited reference (Jacobson et al. (2013)) nor clinical work on human RP patients confirms that patients with only RGCs exist; most probably the INL just thins? Please edit the sentences accordingly.

- p.6, 2nd paragraph: Please edit "Hence, patients with an intact and functional inner nuclear layer could be eligible..."

- remove last sentence "Thus, targeting of ReaChR to retinal ON-BPCs.....", as inconsistent to argument why to use RGCs and since targeting to ON-BPCs (at least in the mouse) has been achieved previously.

- Why has AAV-targeting of ON-BPCs not been accomplished in primates? Has it never been tried or has it not worked? This negative result should be discussed as it is an important translational issue and affirmed by respective references and/or own data. In particular, as you show strong expression of AAV8-2YF after 3-4 days in the macaque explant, I would like to see 1 macaque explant transduced with ReaChR under the GRM6/sv40, just for feasibility. This would add important translational information. Under these lines, please add the transduction figure of the "Reply to reviewers" to the manuscript; it contains important information.

- The argument to try to achieve with optogenetics what retinal implants do is understandable, but why not try to do better and more specific if you have the tools for it, i.e. go for ON-BPCs.

I apologize for insisting on this, but I don't see the rationale of the authors and it appears to me as if molecular tools were chosen somehow arbitrarily.

(5) I am unable to follow the authors reply to this point.

- "The goal of this experiment was to show that the same promoter (hSyn) is functional in mouse and human RGCs". But hSyn is an ubiquitous neuronal promoter? Why should this be the goal? As you achieved nice expression with AAV8-2YF in macaque in 3 days only, why not use AAV in human explants as expression should be similarly fast. As you state yourself, latter has not yet been achieved and would be an important step towards translation, particularly as you also mention yourself that lentivirus due to its bad retinal penetration will not be useful in the clinic. I would

really like to see an AAV-transduced human retina.

- you state that you used a stronger and faster promoter in macaque (CAG), but hSyn should be equally strong and fast as CAG, as also ubiquitous (for neurons at least). To my knowledge, expression speed with AAV is mainly due to the AAV capsid/type(ss or sc). Did you try hSyn? If yes, it would help if you would include photomicrographs of CAG and hSyn in comparison to clarify your statement, or at least references of work who investigated this, if available.

(7) Behavioral paradigms

Although the authors complemented their behavioral data with the light/dark box tests, their data again only gives information about responses to simple light ON-OFF stimuli. Whilst having the behavioral setting in place, the authors should have used the elegantly modified box and improved protocols developed by Cehajic-Kapetanovic J et al. (2015, *Curr Biol* 25) to tease out what treated mice can actually see, i.e. temporal resolution, spatial acuity, natural scenes, in comparison to WT mice. As the authors compare their approach with retinal implant, such information seems crucial in order to determine how the optogenetic approach compares to the existing electrical therapy.

Discussion (p.6/7):

- 3rd paragraph: Please remove the 4 sentences as defocusing and misleading: "When rhodopsin is used as an optogenetic tool in bipolar cells or RGCs,.....it remains questionable if bipolar cells or RGCs would have access to this biochemical pathway of chromophore replenishment.

- Please edit the last sentence of the 3rd paragraph to the following: "In contrast, microbial opsins, melanopsin and Opto-mGluR6 have the advantage to not bleach because their retinal stays covalently bound"

- - 4th paragraph: "We think that visual acuity.... Would be best studied with behavioral tests in AAV injected non-human primates as a next step". But this is by far more ethically problematic, much more costly and would not give much more information than performing visual performance tests in mice. See above point.

Minor points:

- the references don't appear homogenous in the text (1 versus 2 authors et al.)

- you state you are the first to transduce macaque ex vivo with an AAV - but is in vivo (that has been done) not more translational? I think this argument should be removed.

In brief, from my point of view, the novelty/innovativeness/quality and robustness of this work is insufficient for publication in a renowned journal such as *EMBO Mol Med*. This does not mean that I would not like to see this work published elsewhere as it contributes experimental experiences to potential optogenetic vision recovery in the future.

The argumentation of the authors throughout the paper casts a somehow misleading view on the present status of optogenetic vision recovery and the directions and hurdles of translatability, which lie clearly on the molecular side, i.e. finding suitable viral tools and promoters for macaque and human tissue; optogenetic tools in terms of light intensity are already available with properties advantageous of ReaChR.

As the authors have 3 systems for parallel translational investigations available, a strength of the paper, the authors should have used the identical promoter (hSyn) and AAV8(2YF), which they showed expresses rapidly in the macaque, also in human tissue. At present, the chosen molecular approaches and argumentations seems inconsistent and somehow suboptimal.

Also, since the authors justify targeting retinal ganglion cells (and not bipolar cells) to compare with vision recovered by retinal implants, I expected from the authors to investigate the ReaChR-recovered vision (in the mouse) and compare the results to published data from implants. Such data is, despite requests from myself and reviewer 3, still missing after revision.

2nd Revision - authors' response

14 June 2016

Referee #1 (Comments on Novelty/Model System):

This manuscript has been very well revised. It was a pleasure to read. It is a very important contribution to our knowledge of optogenetic therapy of blindness, and the findings have major clinical implications.

We appreciate the reviewer's comments and the opportunity to have the manuscript revised.

Referee #2 (Comments on Novelty/Model System):

The only novelty of this work is MEA recordings from one human retinal explant expressing an optogenetic activator. While it is nice to see that cultured human retinas seem to be viable for some days, no new insight on the translatability of the molecular tools (such as viruses and Promoters) is given in this work nor any new insights on the goodness of optogenetic retinal Ganglion cell targeted vision recovery compared to retinal implants.

For our revised manuscript, we have performed additional experiments with human retinal explants using an AAV-vector, showing ReaChR triggered light responses.

We hope that the reviewer will appreciate that the main goal was to demonstrate that red-shifted channelrhodopsin stimulation could be a promising new therapeutic strategy to restore light responses in the blind human retina (in addition to macaque and mice) at light intensities considered safe per regulatory guidelines on artificial radiation.

Referee #2 (Remarks):

Thank you for the extensive explanations and revisions. Unfortunately some points still remain that I would like to see addressed. I refer to the points of the initial review:

(1),(d),(4): Why to use ReaChR and what cells to target (paragraph 2 Discussion)

- The authors argue that RGCs are the only target for patients with highly disorganized or missing bipolar cell layer. However, neither the cited reference (Jacobson et al. (2013)) nor clinical work on human RP patients confirms that patients with only RGCs exist; most probably the INL just thins?

We have now cited a recent study (Jones et al. Retinal remodeling in human retinitis pigmentosa. *Experimental Eye Research*, 2016) showing that there are indeed human patients who exhibit severe remodeling defects leading to highly disorganized bipolar cell layer: Please see (Jones et al. 2016) page 12: "These changes are likely creating corruptive circuitry in nominally, bipolar cell populations through alterations in the connectivities of dendritic trees and supernumerary axons." Please see also (Jones et al. 2016) page 14 (Chapter 4.10. Optogenetics): "Targeting of ganglion cells in late-stage degeneration is appealing in those patients where bipolar or amacrine cell death and remodeling may render upstream targets unavailable.

Please edit the sentences accordingly.

- p.6, 2nd paragraph: Please edit "Hence, patients with an intact and functional inner nuclear layer could be eligible..."

- remove last sentence "Thus, targeting of ReaChR to retinal ON-BPCs.....", as inconsistent to argument why to use RGCs and since targeting to ON-BPCs (at least in the mouse) has been achieved previously.

We have removed this last sentence.

- Why has AAV-targeting of ON-BPCs not been accomplished in primates? Has it never been tried or has it not worked? This negative result should be discussed as it is an important translational issue and affirmed by respective references and/or own data. In particular, as you show strong expression of AAV8-2YF after 3-4 days in the macaque explant, I would like to see 1 macaque explant transduced with ReaChR under the GRM6/sv40, just for feasibility. This would add important translational information. Under these lines, please add the transduction figure of the "Reply to reviewers" to the manuscript; it contains important information.

- The argument to try to achieve with optogenetics what retinal implants do is understandable, but why not try to do better and more specific if you have the tools for it, i.e. go for ON-BPCs. I apologize for insisting on this, but I don't see the rationale of the authors and it appears to me as if molecular tools were chosen somehow arbitrarily.

We think that targeting of the bipolar cells in the macaque retina is beyond the scope of the current study.

We agree that targeting of the bipolar cells is an attractive strategy among other valuable therapy options, depending on the remaining cell-types in blind patients.

(5) I am unable to follow the authors reply to this point.

- "The goal of this experiment was to show that the same promoter (hSyn) is functional in mouse and human RGCs". But hSyn is an ubiquitous neuronal promoter? Why should this be the goal? As you achieved nice expression with AAV8-2YF in macaque in 3 days only, why not use AAV in human explants as expression should be similarly fast. As you state yourself, latter has not yet been achieved and would be an important step towards translation, particularly as you also mention yourself that lentivirus due to its bad retinal penetration will not be useful in the clinic. I would really like to see an AAV-transduced human retina.

Responding to the criticism that we did not use the AAV on the human explant, we have now performed additional experiments in the human retina with the same tool we used for recording ReaChR responses in the macaque retina. In an explant prepared from the peripheral human retina we recorded light responses after 12 days of AAV:CAG-ReaChR incubation (please see new Figure 8).

Please note that human retinal explants are difficult to maintain viable for physiological recording in culture as long as 12 days and we have previously not recorded from explants incubated more than 5 days. We have provided a time course for the expression of the AAV:CAG ReaChR construct in the human retina which indicates that the protein was not detectable until at least 9 days in the culture setting.

For comparison, we also transduced the human retina with the AAV2:hSyn-ReaChR construct. At day 9, the time course shows that the expression of the AAV2:hSyn-ReaChR construct was much weaker, compared to AAV:CAG ReaChR. We attempted to but could not record light responses from human retina transduced with the AAV2:hSyn-ReaChR construct.

- you state that you used a stronger and faster promoter in macaque (CAG), but hSyn should be equally strong and fast as CAG, as also ubiquitous (for neurons at least). To my knowledge, expression speed with AAV is mainly due to the AAV capsid/type(ss or sc). Did you try hSyn? If yes, it would help if you would include photomicrographs of CAG and hSyn in comparison to clarify your statement, or at least references of work who investigated this, if available.

Please see Fig 5A for AAV2: hSyn:ReaChR-mCIt transduced macaque retina at 10 days post incubation.

(7) Behavioral paradigms

Although the authors complemented their behavioral data with the light/dark box tests, their data again only gives information about responses to simple light ON-OFF stimuli. Whilst having the behavioral setting in place, the authors should have used the elegantly modified box and improved protocols developed by Cehajic-Kapetanovic J et al. (2015, Curr Biol 25) to tease out what treated mice can actually see, i.e. temporal resolution, spatial acuity, natural scenes, in comparison to WT mice. As the authors compare their approach with retinal implant, such information seems crucial in order to determine how the optogenetic approach compares to the existing electrical therapy.

Following the editor's advice, we have not performed additional behavioral tests.

Discussion (p.6/7):

- 3rd paragraph: Please remove the 4 sentences as defocusing and misleading: "When rhodopsin is used as an optogenetic tool in bipolar cells or RGCs,.....it remains questionable if bipolar cells or RGCS would have access to this biochemical pathway of chromophore replenishment.

We agree that rhodopsin is a highly valuable alternative to microbial opsins. Nevertheless, we think that bleaching of rhodopsin is an important point that has to be addressed in the discussion. We have

now rephrased the last sentence: “As Müller glial cells have been identified as an alternative source of retinal recycling for cone photoreceptors (Kaylor, Yuan et al., 2013), it would be interesting to investigate if bipolar cells or RGCs have access to this biochemical pathway of chromophore replenishment.”

- Please edit the last sentence of the 3rd paragraph to the following: "In contrast, microbial opsins, melanopsin and Opto-mGluR6 have the advantage to not bleach because their retinal stays covalently bound"

We have now added the following sentence: “Melanopsin and Opto-mGluR6 have also the advantage that they are resistant to bleaching (Sexton, Golczak et al., 2012, van Wyk et al., 2015).”

-- 4th paragraph: "We think that visual acuity.... Would be best studied with behavioral tests in AAV injected non-human primates as a next step". But this is by far more ethically problematic, much more costly and would not give much more information than performing visual performance tests in mice. See above point.

We believe that the tests of higher order visual function and visually guided cognitive behavior would be best studied with *in vivo* injected non-human primates for meaningful insight in to the type of visual function optogenetics could restore in blind patients.

Minor points:

- the references don't appear homogenous in the text (1 versus 2 authors et al.)

We used the EMBO J formatting style (EndNoteX7). We would be happy to provide changes of the reference format, if requested.

- you state you are the first to transduce macaque ex vivo with an AAV - but is in vivo (that has been done) not more translational? I think this argument should be removed.

We have removed this argument.

In brief, from my point of view, the novelty/innovativeness/quality and robustness of this work in insufficient for publication in a renowned journal such as EMBO Mol Med. This does not mean that I would not like to see this work published elsewhere as it contributes experimental experiences to potential optogenetic vision recovery in the future.

The argumentation of the authors throughout the paper casts a somehow misleading view on the present status of optogenetic vision recovery and the directions and hurdles of translatability, which lie clearly on the molecular side, i.e. finding suitable viral tools and promoters for macaque and human tissue; optogenetic tools in terms of light intensity are already available with properties advantageous of ReaChR.

As the authors have 3 systems for parallel translational investigations available, a strength of the paper, the authors should have used the identical promoter (hSyn) and AAV8(2YF), which they showed expresses rapidly in the macaque, also in human tissue. At present, the chosen molecular approaches and argumentations seems inconsistent and somehow suboptimal.

Also, since the authors justify targeting retinal ganglion cells (and not bipolar cells) to compare with vision recovered by retinal implants, I expected from the authors to investigate the ReaChR-recovered vision (in the mouse) and compare the results to published data from implants. Such data is, despite requests from myself and reviewer 3, still missing after revision.

We would like to stress that the prime objective of our study was not the comparison of viral tools or promoters. Rather, what our work highlights is that red-shifted channelrhodopsins in general could be a promising new therapeutic strategy to restore light responses in the blind human retina at safe light intensities.

Thank you for the submission of your revised manuscript to EMBO Molecular Medicine and apologies for the delay due also to the fact that I have been on work-related travel for the past two weeks.

We have now received the enclosed report from Reviewer 2, who was asked to re-assess your manuscript. As you will see s/he is now globally supportive although there are a few remaining issues. Please amend the text to accommodate these requests. I will be making an Editorial decision on your final, revised version, provide the issues raised are dealt with as required.

Please also consider the following final Editorial amendments/requests:

- 1) You refer to the movie files as "Expanded View Movie Figure 1" and the EV movie legends are still within the article. Please rename the movies to "Expanded View Movie 1" ... and move their legends from the article into to individual read-me files > individually zipped together with the related movie files (file name "EMM5699_MovieEV1.zip"...)
- 2) Please provide a short stand first text (maximum of 300 characters, including space) as well as 2-5 one sentence bullet points that summarise the paper. Please write the bullet points to summarise the key NEW findings. They should be designed to be complementary to the abstract - i.e. not repeat the same text. We encourage inclusion of key acronyms and quantitative information (maximum of 30 words / bullet point). Please use the passive voice and attach these in a separate Word-file or send them by email. We will incorporate them accordingly.
- 3) You are also welcome to suggest a striking image or visual abstract to illustrate your article. If you do please provide a jpeg file 550 px-wide x 400-px high.
- 4) Please include the section "Author contributions" within the article file.

Please submit your revised manuscript within two weeks. I look forward to seeing a revised form of your manuscript soon, and possibly no later than two weeks.

REFeree REPORT

Referee #2 (Remarks): Thank you for the thorough revisions, the text has been improved in fluency and the information given is more consistent. I also appreciate the authors' efforts to record from an AAV transduced human retina, congratulations!

The text has been generally tuned down, but please also revise the Abstract and the "Paper Explained" sections under these lines; they currently sound like you developed ReaChR.

Recommendations for the "Paper Explained" section:

- 1) Last sentence of "PROBLEM": Change to i.e. "Recently, a red-shifted ChR2 has been developed, which we tested for a safe therapeutic approach to recover vision".
- 2) "IMPACT" (much exaggerated). Please just write what you did, i.e.: "We showed in our translational approach that red-shifted ChR2s are capable of triggering retinal light responses under safe illumination intensities, not only in mouse, but also in macaque and human ex vivo retinas."

Recommendations for the Abstract:

- 1) ReaChR is not "novel"
- 2) orange light levels are not "well below" the safety limit, there are just "below" the limit

Besides above, I only see minor additional revisions:

- (1) Please state in the text clearly for each section (mouse, macaque, human) (a) what AAV capsid has been used and (b) what light intensity has been used.
- (2) Discussion: Briefly discuss the potential consequences (for vision) of only recovering RGC ON-responses.
- (3) Recommendations to improve Abstract: blue ChR2 phototoxic, at 600 nm one could use 1000x dimmer light; ReaChR was shown to be activated at 600nm. Thus we tested if this holds true in the

retina using 3 models in parallel: mouse, macaque and human. We found that safe orange illumination is able to trigger visual responses in all three systems. We also show for the first time optogenetically induced spikes in human retina.

(4) p.8, line3: why should results in macaque in vivo be improved? Please explain briefly.

(5) Methods p.12: Do I understand correctly, all hSyn constructs are in AAV2 and all CAG constructs in AAV8 (please clarify)? If yes, do you not see this to be the reason why in Expanded View Fig. 4 hSyn is not visible after 9 days? I would suggest that it is the slower expression of AAV2 compared to AAV8 that makes hSyn seem weaker (although viral promoters are generally also strong....). Also, p.4 last paragraph to induce "stronger" expression you used CAG I suppose, not for "faster" expression.

3rd Revision - authors' response

09 August 2016

REVISION # 3 point by point author responses

Referee #2 (Remarks):

Thank you for the thorough revisions, the text has been improved in fluency and the information given is more consistent. I also appreciate the authors' efforts to record from an AAV transduced human retina, congratulations!

We are glad to provide thorough revisions to enhance our paper and appreciate his/her comments on the delicate experiments we performed on human retina.

The text has been generally tuned down, but please also revise the Abstract and the "Paper Explained" sections under these lines; they currently sound like you developed ReaChR.

Recommendations for the "Paper Explained" section:

- 1) Last sentence of "PROBLEM": Change to i.e. "Recently, a red-shifted ChR2 has been developed, which we tested for a safe therapeutic approach to recover vision".*
- 2) "IMPACT" (much exaggerated). Please just write what you did, i.e.: "We showed in our translational approach that red-shifted ChR2s are capable of triggering retinal light responses under safe illumination intensities, not only in mouse, but also in macaque and human ex vivo retinas."*

We have revised our "Paper Explained" section, based on these recommendations.

Recommendations for the Abstract:

- 1) ReaChR is not "novel"*
- 2) orange light levels are not "well below" the safety limit, there are just "below" the limit*

We have revised our abstract.

Besides above, I only see minor additional revisions:

- (1) Please state in the text clearly for each section (mouse, macaque, human) (a) what AAV capsid has been used and (b) what light intensity has been used.*

We have now added more explicit references as to what capsid was used for each experiment, including in the figure legends.

- (2) Discussion: Briefly discuss the potential consequences (for vision) of only recovering RGC ON-responses.*

We have now elaborated our discussion of this topic in the Discussion section, p.7 para 2.

- (3) Recommendations to improve Abstract: blue ChR2 phototoxic, at 600 nm one could use 1000x dimmer light; ReaChR was shown to be activated at 600nm. Thus we tested if this holds true in the retina using 3 models in parallel: mouse, macaque and human. We found that safe orange*

illumination is able to trigger visual responses in all three systems. We also show for the first time optogenetically induced spikes in human retina.

We have revised our abstract.

(4) p.8, line3: why should results in macaque in vivo be improved? Please explain briefly.

We expect that retinæ isolated from *in vivo* injected macaques would contain a much higher number of ReaChR-transduced RGCs and the sensitivity of the optogenetic light responses from freshly isolated retinæ would be higher than ones that have been sitting in an incubator for a number of days.

(5) Methods p.12: Do I understand correctly, all hSyn constructs are in AAV2 and all CAG constructs in AAV8 (please clarify)? If yes, do you not see this to be the reason why in Expanded View Fig. 4 hSyn is not visible after 9 days? I would suggest that it is the slower expression of AAV2 compared to AAV8 that makes hSyn seem weaker (although viral promoters are generally also strong....). Also, p.4 last paragraph to induce "stronger" expression you used CAG I suppose, not for "faster" expression.

Yes, indeed all the AAV experiments with the hSyn promoter were performed using the AAV2 capsid vector and all the CAG AAV constructs were in AAV8. This could have contributed to the slower expression of ReaChR under the hSyn promoter.

We have added a statement p.6, last para noting that both the promoter as well as capsid could have played a role in the expression of the CAG ReaChR AAV construct *ex vivo*.

Corresponding Author Name: Jens Duebel
 Journal Submitted to: EMBO Molecular Medicine
 Manuscript Number: EMM-2015-05699